


# 1 Spatial reconstruction of long-term (2003-2020) sea surface $p$CO$_2$ in the South China Sea using a machine learning based regression method aided by empirical orthogonal function analysis

Zhixuan Wang[1], Guizhi Wang[1,2], Xianghui Guo[1], Yan Bai[3], Yi Xu[1] and Minhan Dai[1,*]
[1]State Key Laboratory of Marine Environmental Science and College of Ocean and Earth Sciences, Xiamen University, Xiamen,
361102, China
[2]Fujian Provincial Key Laboratory for Coastal Ecology and Environmental Studies, Xiamen University, Xiamen, 361102, China
[3]State Key Laboratory of Satellite Ocean Environment Dynamics, Second Institute of Oceanography, State Oceanic
Administration, Hangzhou, 310012, China
*Correspondence to*: Minhan Dai (mdai@xmu.edu.cn)
**Abstract.** The South China Sea (SCS), the largest marginal sea of the North Pacific Ocean, is one of the world's most studied
model ocean margins in terms of its carbon cycle, where intensive field observations including sea-surface carbon dioxide partial
pressure ($p$CO$_2$) have been conducted over the last two decades. However, the datasets of cruise-based sea surface $p$CO$_2$ are still
temporally and spatially incomplete. Using a machine learning-based method facilitated by empirical orthogonal function (EOF)
analysis capable of constraining the spatiality, this study provides a reconstructed dataset of the monthly sea surface $p$CO$_2$ in the
SCS with a reasonably high spatial resolution (0.05º×0.05º) and temporal coverage between 2003 and 2020. We validate our
reconstruction with three independent testing datasets where, TEST.1 includes 10% of our observed data, TEST.2 includes four
independent underway datasets corresponding to four seasons, and TEST.3 includes a continuous observed dataset from 2003 –
2019 at the South East Asia Time-Series (SEATs) station located in the northern basin of the SCS. Our TEST.1validation
demonstrated that the reconstructed $p$CO$_2$ field successfully simulated the spatial and temporal patterns of sea surface $p$CO$_2$. The
root-mean-square error (RMSE) between our reconstructions and observed data in TEST.1 averaged to ~10 µatm, which is much
smaller (by ~50%) than that between the remote sensing (RS) and observed data. TEST.2 verified the accuracy of our
reconstruction model in data months lacking observations, showing a near-zero bias (RMSE: ~8 µatm). TEST.3 tested the
accuracy of the reconstructed long-term trend, showing that at the SEATs Station, the difference between the reconstructed $p$CO$_2$
and observations ranged from -10 to 4 µatm (-2.5 to 1%). In addition to the typical machine learning performance metrics, we
present a new method to assess the uncertainty that includes the bias from the reconstruction and its sensitivity to the features, and
successfully quantifies the spatial distribution patterns of uncertainty. These validations and uncertainty analysis strongly suggest
that our reconstruction is effectively captures the main features of both the spatial and temporal patterns of sea surface $p$CO$_2$ in the



SCS. Using the reconstructed dataset, we show the long-term trends of sea surface $p\mathrm{CO}_2$ in 5 sub-regions of the SCS with
differing physico-biogeochemical characteristics. We show that mesoscale processes such as the Pearl River plume and China
Coastal Currents significantly impact sea surface $p\mathrm{CO}_2$ in the SCS during different seasons. While the SCS is overall a weak
source of atmospheric $\mathrm{CO}_2$, the northern SCS acts as a sink, showing a trend of increasing strength over the past two decades.
Key words: Sea surface $p\mathrm{CO}_2$; reconstruction; machine learning; South China Sea

**1 Introduction**
The ocean possesses much of the global capacity for atmospheric carbon dioxide ($\mathrm{CO}_2$) sequestration and annually mitigates
22–26% of the anthropogenic $\mathrm{CO}_2$ emissions associated with fossil fuel burning and land use change during the period 1960–2019
(Friedlingstein et al., 2020). However, it remains largely unknown whether and by how much the ocean will continue to act as a
sink for anthropogenic $\mathrm{CO}_2$, i.e., the extent of its climate change mitigation capacity to understand climate-carbon coupled
systems and develop zero-emission strategies and actions. Ocean margins, an essential part of the land-ocean continuum,
contributed ~ 10-20% of the global ocean $\mathrm{CO}_2$ sequestration with only 7% of the surface area and represent a particularly
challenging regime (e.g., Chen and Borges, 2009; Dai et al. 2022; Laruelle et al., 2014). This is primarily attributed to the ocean
margins' extremely complex and dynamic processes, often characterized by large spatial and temporal variability of air-sea $\mathrm{CO}_2$
fluxes that lead to even larger uncertainty in their prediction than those occurring in the open ocean (Dai et al., 2013, 2022; Cao et
al., 2020; Laruelle et al., 2014; Chen and Borges, 2009 and the references therein). Limited spatiotemporal coverage of
observational data is an important source of these uncertainties.
In recent years, many studies use numerical models or data-based approaches to improve estimates of sea surface $\mathrm{CO}_2$ distribution
and the accuracy of the global carbon budget for periods and regions with poor coverage of observational data (Rödenbeck et al.,
2015; Wanninkhof et al., 2013). Numerical ocean models of performance can successfully quantify the generally increasing trend
in oceanic $\mathrm{CO}_2$ and some critical processes of carbon cycling (e.g., net ecosystem production), but still suffer from regional and
seasonal differences in their estimates of the ocean carbonate system (Luo et al., 2015; Mongwe et al., 2016; Tahata et al., 2015;
Wanninkhof et al., 2013). Thus, data-based approaches have become a popular alternative to biogeochemical models (Jones et al.,
2014; Lefèvre et al., 2005;Landschützer et al., 2014, 2017; Telszewski et al., 2009). The former typically use statistical
interpolations and regression methods. Statistical interpolations improve the spatial coverage of observational data, but do not
work for the period without observational data. Regression methods allow mapping of the relationship between the observed
carbon dioxide partial pressure ($p\mathrm{CO}_2$) data and other parameters that may drive changes in surface ocean $p\mathrm{CO}_2$, and then
extrapolation of this relationship to improve estimates of the spatiotemporal distribution of $p\mathrm{CO}_2$. The development of machine
learning methods and remote sensing-derived products (as proxy variables in regression methods) have aided the development of
data-based methods (Rödenbeck et al., 2015; Bakker et al., 2016) which, with spatial-temporal standardization, can improve the



model results of the oceanic carbonate system by numerical assimilation methods. However, because of the complex and dynamic
nature of biogeochemical and physical processes in coastal areas, characterization of sea surface $p\text{CO}_2$ and subsequently the
air-sea $\text{CO}_2$ fluxes both in time and space in marginal seas remains challenging.
The South China Sea (SCS) is the largest marginal sea of the North Pacific Ocean with a surface area of $3.5\times10^6$ km$^2$. Although
extensive field observations have been conducted of sea surface $p\text{CO}_2$ in the SCS in the past two decades, their spatial and
temporal coverage is still limited in different physical-biogeochemical domains of the SCS and at sub-seasonal time scales (e.g.,
Guo et al., 2015; Li et al., 2020; Zhai et al., 2005; Zhai et al., 2013). Therefore, there is a clear need to achieve surface water $p\text{CO}_2$
coverage in the SCS with a highest spatiotemporal resolution as possible with the aim to better estimate sea surface $p\text{CO}_2$ and thus
air-sea $\text{CO}_2$ fluxes in the SCS and help develop improved initial conditions of numerical models. Moreover, a reasonably high
spatiotemporal resolution of $p\text{CO}_2$ data can help identify the controlling factors of $p\text{CO}_2$ changes in the SCS, and reliably resolve
the long-term changes.
Zhu et al. (2009) presented an empirical approach that estimated sea surface $p\text{CO}_2$ in the northern SCS in summer using
satellite-derived data (sea surface temperature, SST; chlorophyll *a*, Chl *a*), and their validation results show that the
reconstructed $p\text{CO}_2$ data was generally consistent with the underway observed data. However, it should be noted that the large
uncertainty of estimates from their study was caused by the limited underway observed data from only two summer cruises (July
2000 data were used for algorithm tuning and those of July 2000 for validation). Jo et al. (2012) developed a neural
networking-based algorithm by using SST and Chl *a* to estimate sea surface $p\text{CO}_2$ in the northern SCS. Sea surface $p\text{CO}_2$ data in
this study were collected from May 2001, and, February, July 2004. The difference between the reconstructed $p\text{CO}_2$ data of Jo et
al. (2012) and the observed data reflects a relatively large bias (the resultant RMSE (root-mean-square error) falls in the range
32.6 to 44.5 µatm, reported in Wang et al., 2021). Bai et al. (2015) used a 'mechanic semi-analytical algorithm' to estimate the
satellite remote sensing-derived sea surface $p\text{CO}_2$ data in the East China Sea during 2000–2014, and then used this algorithm to
estimate sea surface $p\text{CO}_2$ for the whole China Sea. These authors also pointed the limitation of their mechanistic
semi-analytical algorithm (MeSAA) which did not fully account for some local processes and therefore causes errors (the RMSE
is about 45 µatm in the SCS (reported in Wang et al., 2021). Bai et al. (unpublished) subsequently used a machine learning based
non-linear regression to develop a retrieval algorithm for seawater $p\text{CO}_2$ in the China Sea, and the satellite-derived $p\text{CO}_2$ data
from 2003-2018 were provided by the SatCO$_2$ platform (www.SatCO2.com) with a RMSE of 21.1 µatm in China Seas.
To take advantages of both the high spatiotemporal resolution of the remote sensing-derived $p\text{CO}_2$ data (RSdata) and the accuracy
of the observational data, Wang et al. (2021) reconstructed the basin–scale sea surface $p\text{CO}_2$ in the SCS in summer by using the
empirical orthogonal function (EOF) based on a multi-linear regression method and demonstrated the reliability of the
reconstructions. However, when the spatial standard deviation of observed data is relatively large because of the influence of
outliers, the reconstruction results may be biased (Wang et al., 2021). Therefore, many studies used the machine learning-based



regression method to reduce the influence of outliers for open ocean areas, with a RMSE of <17 µatm in most cases (e.g., Zeng et
al., 2017; Li et al., 2019).
Building upon the EOF method that significantly improved the reconstruction in terms of spatial pattern and accuracy (Wang et al.,
2021), we developed a machine learning-based regression method facilitated by the EOF to fully resolve the long-term spatial
distribution of sea surface $p$CO$_2$ at a resolution of 0.05º×0.05º in the SCS. In addition to the typical machine learning performance
metrics, we present a novel uncertainty calculation method that incorporates the bias of both the reconstruction and the sensitivity
of reconstructed models.

**2 Study site and data sources**
**2.1 Study area**
The SCS, located in the western Pacific, has a maximum water depth of ca. 4700 m (e.g., Gan et al., 2006, 2010). The rhombus
deep-water basin with a southwest-northeast direction accounts for about half of the total area of the SCS (Fig. 1). The
oceanography of the SCS is largely modulated by the Asian monsoon and the topography, thus exhibiting seasonally varying
surface circulation, river inputs, and upwelling. Forced by the northeast winds in winter, the circulation of the upper layer shows a
large cyclonic circulation structure (red solid line in Fig. 1), while in summer it exhibits an anticyclonic circulation structure
forced by southwest winds (red dashed line in Fig. 1; Hu et al. 2010). In the northern SCS, the Pearl River discharges into the SCS
with an annual freshwater input of $3.26 \times 10^{11}$ m$^3$ (e.g., Dong et al., 2004; Dai et al., 2014). The area influenced by the Pearl River
Plume may extend southeastward to a few hundred kilometers from the river estuary in summer because of the monsoon wind
stress (Dai et al., 2014). The northern and western coastal regions of the SCS also feature summer coastal upwelling in summer,
mainly including the Eastern Guangdong and Qiongdong upwelling systems in the northern SCS and the Vietnam upwelling
systems in the western SCS (e.g., Cao et al., 2011; Chen et al., 2012; Gan et al., 2006; Gan et al., 2010; Li et al., 2020).

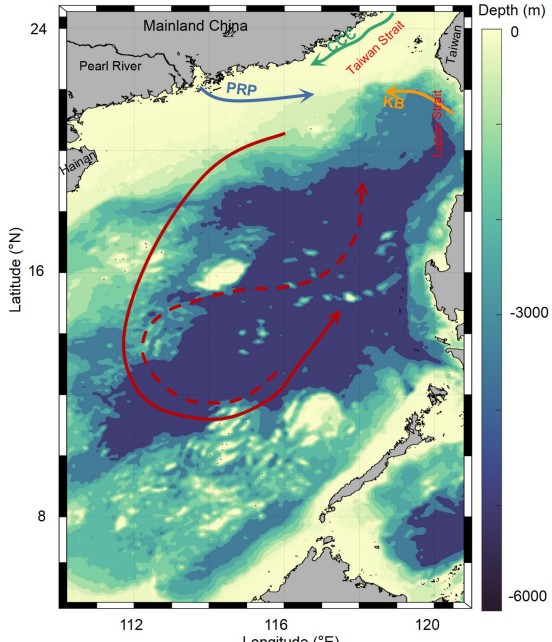


**Figure 1. Topographic map of the South China Sea (SCS) showing a basin wide cyclonic gyre in winter (solid line) and an anticyclonic gyre over the southern half of the SCS in summer (dashed line). Also shown are the Kuroshio Branch (KB, orange line), the China Coastal Current (CCC, green line), and the Pearl River plume (PRP, blue line).**

The SCS is a semi-enclosed sea basin with dynamic water exchange with the East China Sea via the Taiwan Strait and Western Pacific via the Luzon Strait (Fig. 1). In winter, driven by the winter monsoon, the China Coastal Current (CCC, yellow line in Fig. 1; Han et al., 2013; Yang et al., 2022) flows south along the Chinese mainland through the Taiwan Strait, and occupies the northern SCS with cold, fresh, nutrient-rich waters. The strong northeast winds in winter also slow down the western boundary ocean current, forcing the intrusion of Kuroshio water, which shows high surface salinity and high total alkalinity, into the SCS via the Luzon Strait (orange line in Fig. 1; Du et al., 2013; Park, 2013; Yang et al., 2022).

**2.2 Observational $p$CO$_2$ data**

Data collected from field surveys during the study period 2003-2020 are summarized in Table 1. Most observations were made in July, and fewer observations were made in March and December of each year. The rough sea state in the SCS in winter and early spring limited the survey during these seasons. Data collected from July 2000 to January 2018 were originally published by Li et al. (2020). The in situ $p$CO$_2$ were collected from R/Vs *Dongfanghong-2, Tan Kah Kee (TKK)*, etc. (shown in Table 1). The data collection methods used in this study have been introduced in Li et al. (2020). The spatial coverage and frequency of the observations are shown in Figure 2 and show that there are pronounced seasonal changes and that the data cover a large spatial area. For example, the spatial coverage of the observed data in spring and fall are relatively uniformly distributed, and the south end of the spatial coverage reaches 5 ºN in spring, whereas that during other seasons is concentrated in the northern and central





regions of the SCS. In addition, only one observation was made in the basin area in winter, while the northern coastal area was
more frequently surveyed, especially in summer.
**Table 1. Summary of the seasonal observational data of sea surface $p$CO$_2$ in the South China Sea for the period 2003-2020**
**used in this study.**

| Season | Spring | | | Summer | | |
|---|---|---|---|---|---|---|
| | March | April | May | June | July | August |
| Cruise time | 2004.03 | 2005.04 2008.04 2009.04 2012.04 2020.04 | 2004.05 2011.05 2014.05 2020.05 | 2006.06 2016.06 2017.06 2019.06 2020.06 | 2004.07 2005.07 2007.07 2008.07 2009.07 2012.07 2015.07 2019.07 | 2007.08 2008.08 2019.08 |
| Season | Fall | | | Winter | | |
| | September | October | November | December | January | February |
| Cruise time | 2004.09 2007.09 2008.09 2020.09 | 2003.10 2006.10 | 2006.11 2010.11 | 2006.12 | 2009.01 2010.01 2018.01 | 2004.02 2006.02 |

**\*Data were collected before February 2018 except those collected in July 2015 and June 2017 which are from Li et al.**
**(2020).**

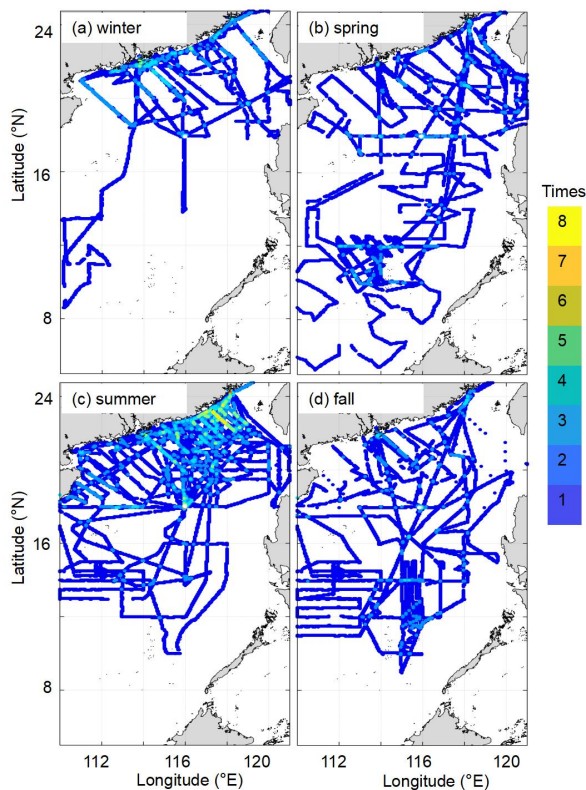


**Figure 2. Cruise tracks of the observations conducted in the South China Sea in each season from 2000 to 2020: (a) winter, (b) spring, (c) summer, and (d) fall. The data were collected before February 2018 except those collected in July 2015, June 2017 which are from Li et al. (2020).**


Figure 3 shows the spatial and temporal distributions of surface water $pCO_2$. Seasonally, the lowest $pCO_2$ occurs in January, and the highest concentrations occur in May and June. Spatially, the $pCO_2$ distribution in the basin is relatively homogeneous, but shows large variability in the northern region. In the northern coastal area in summer, the observed $pCO_2$ distribution is affected by the Pearl River plume (yielding low values) and coastal upwelling (yielding high values), which last into early fall. In winter and early spring, relatively low $pCO_2$ values were determined in the coastal area. In addition, the high $pCO_2$ values recorded on the western side of the Luzon Strait in December demonstrate the influence of winter upwelling during some of the surveys.

In addition to the above observational data, we selected four independent surveys corresponding to four seasons, and a continuous observation dataset during 2003–2019 at the Southeast Asia Time-Series (SEATs) station (Dai et al., 2022) as two important independent testing datasets.

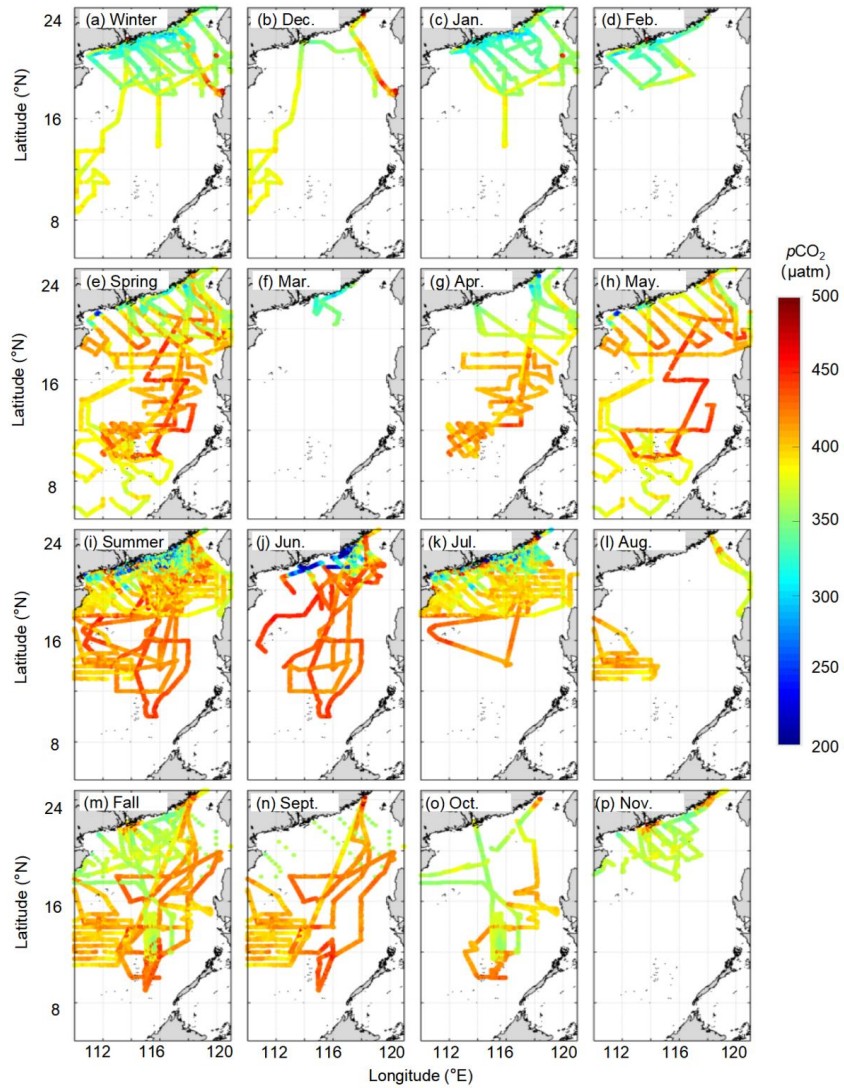


**Figure 3. Seasonal and monthly sea surface $p$CO$_2$ fields in the South China Sea. The data sources can be found in Table 1.**


**2.3 Remote sensing-derived sea surface $p$CO$_2$ data**
The gridded (0.05º×0.05º) remote sensing-derived $p$CO$_2$ data covered almost the entire SCS (5–25º N, 109–122º E), and show the
major CO$_2$ variation at a large scale (Wang et al., 2021; Bai et al., unpublished). Further details of the remote sensing (RS) data
can be found in the SatCO$_2$ platform (www.SatCO2.com).
A grid-to-grid comparison was undertaken (Fig. 4) and the RMSE of the RS data-derived $p$CO$_2$ values were compared with the
observed $p$CO$_2$ data (Table 2). This comparison shows that the difference between the RS and the observed $p$CO$_2$ data ranges from





to 120 µatm in the coastal area, and that the largest biases occur in summer. In terms of the RMSE (Table 2), the largest bias
reaches 30.0 µatm in summer. Bai et al. (2015; unpublished) and Wang et al. (2021) pointed out that relatively large discrepancies
may reflect the limitations of the current algorithm, which considers only biological processes and the turbidity induced by the
Pearl River discharge (characterized by Chl *a* and the remote sensing reflectance at 555 nm (rrs555) and does not take into
account the riverine dissolved inorganic carbon and the input of other substances that may affect $pCO_2$.
Because of the relatively large bias of RS data, their direct use may lead to overestimate $pCO_2$ values in the reconstructed field. In
this study, the EOF method was used to compute the spatial patterns of the RS data to assess the accuracy and spatial distribution
of the reconstruction data as a whole and remove the influence of the limited RS data.

**Table 2. Biases between the seasonal remote sensing (RS)-derived $pCO_2$ data and observed $pCO_2$ data, and between the**
**reconstructed and the observed underway $pCO_2$ data. (unit: µatm; the remote sensing-derived $pCO_2$ data during**
**2003-2019 are from www.SatCO2.com and the source of observed data can be found in Table1. The reconstructed $pCO_2$**
**data are from section 3; all data were gridded into 0.05°\*0.05°; /: no data). MAE = mean absolute error; RMSE = root**
**mean square error; $R^2$ = coefficient of determination; MAPE = mean absolute percentage error.**

|  |  | RS | Training data | Testing data I | Testing data II | Testing data III |
|---|---|---|---|---|---|---|
| Spring | MAE | 9.00 | 2.44 | 4.76 | 1.68 | / |
|  | RMSE | 12.70 | 3.47 | 7.43 | 2.26 | / |
|  | $R^2$ | / | 0.98 | 0.92 | / | / |
|  | MAPE | / | 0.01 | 0.01 | / | / |
| Summer | MAE | 16.75 | 2.48 | 8.46 | 5.73 | / |
|  | RMSE | 29.956 | 3.54 | 14.69 | 15.18 | / |
|  | $R^2$ | / | 0.99 | 0.89 | / | / |
|  | MAPE | / | 0.01 | 0.02 | / | / |
| Fall | MAE | 9.93 | 2.41 | 4.90 | 7.133 | / |
|  | RMSE | 13.08 | 3.39 | 6.85 | 8.94 | / |
|  | $R^2$ | / | 0.98 | 0.92 | / | / |
|  | MAPE | / | 0.01 | 0.01 | / | / |
| Winter | MAE | 9.25 | 2.18 | 5.61 | 11.41 | / |
|  | RMSE | 14.26 | 3.14 | 8.82 | 12.63 | / |
|  | $R^2$ | / | 0.98 | 0.89 | / | / |



| | | | | | | |
|---|---|---|---|---|---|---|
| | MAPE | / | 0.01 | 0.01 | / | / |
| | MAE | 11.95 | 2.41 | 6.30 | 5.27 | 6.19 |
| Annual | RMSE | 20.66 | 3.43 | 10.79 | 11.18 | 8.26 |
| | $R^2$ | / | 0.99 | 0.91 | / | / |
| | MAPE | / | 0.01 | 0.01 | / | / |


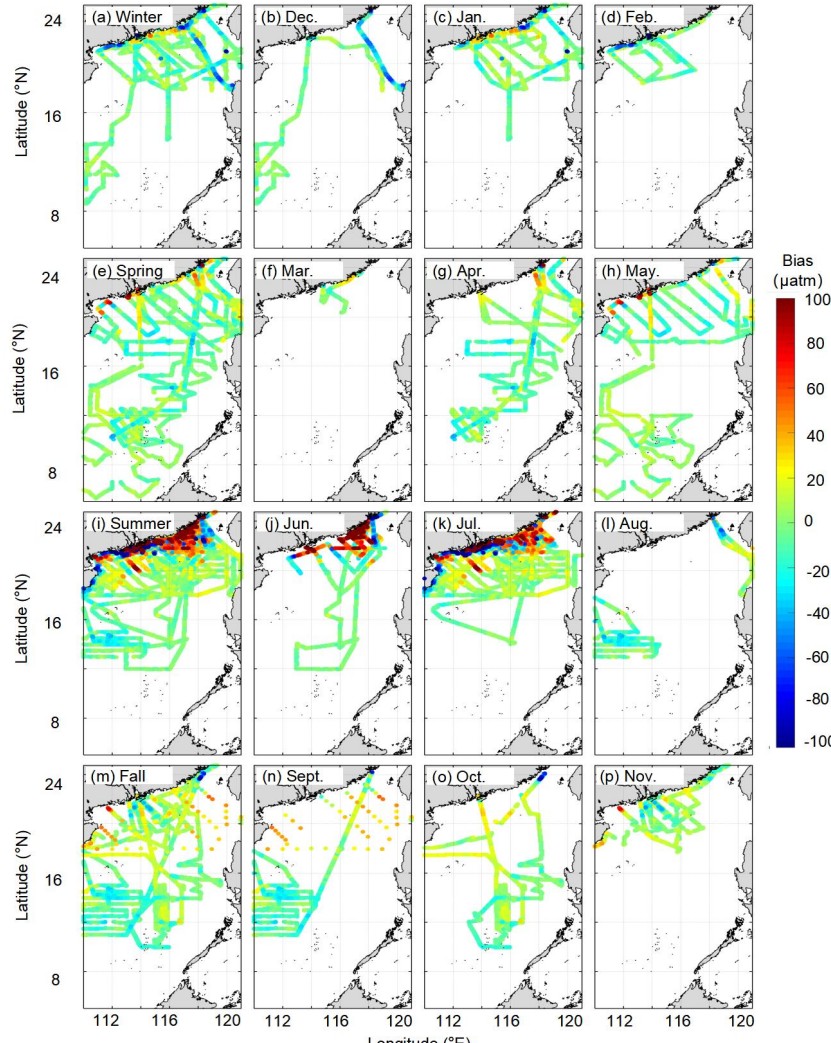


**Figure 4. Differences between the seasonal and monthly remote sensing-derived $p$CO₂ and the observed $p$CO₂ data (the former during 2003-2019 is from www.SatCO2.com, and the source of observed data can be found in Table 1. Both datasets are gridded into 0.05°×0.05°, and the bias is plotted grid by grid).**




**2.4 Other data**

The RS SST data produced by MODIS (https://oceancolor.gsfc.nasa.gov/) are adopted here. The uncertainty of this dataset in the SCS is ~0.27° (Qin et al., 2014). For the SSS data, Wang et al. (in preparation) found a relatively high differential between the values found in different open source databases (i.e., multi-satellite fusion data from https://podaac.jpl.nasa.gov/; model data from https://climatedataguide.ucar.edu/; multidimensional covariance model data from https://resources.marine.copernicus.eu/) and our observed data. Wang et al. (in preparation) reconstructed a SSS database using machine learning methods based on the observation dataset. The bias between the reconstructed SSS and observed data was near-zero (mean absolute error, MAE: ~0.25). Next, we used Chl-$a$ as an indicator of biological influence. Chl-$a$ data from MODIS (https://oceancolor.gsfc.nasa.gov/) are adopted in the present study, which have a bias of ~0.35 on log scale and ~115% in the SCS (Zhang et al., 2006). Atmospheric $pCO_2$ also influences surface water $pCO_2$ through air–sea $CO_2$ exchange. We chose the atmosphere $CO_2$ mole fraction ($xCO_2$) data from the monthly mean $CO_2$ concentrations measured at Mauna Loa Observatory, Hawaii (https://gml.noaa.gov/). The atmospheric $pCO_2$ values were calculated from $xCO_2$ using the method of Li et al. (2020).

**3 Methods**

The $pCO_2$ reconstruction procedure is shown in Figure 5. It includes: (1) data processing and (2) model training and testing. For the former, we first gridded the observed and RS $pCO_2$ data into 0.05°×0.05° grid boxes with monthly temporal resolution. Secondly, we used the $pCO_2$ filling method of Fay et al. (2021) to fill the missing $pCO_2$ measurements with the RS $pCO_2$ data. We then used a feature engineering method to ignore any biases in the data itself or from the $pCO_2$ filling method. Thirdly, the gridded observed $pCO_2$ data and their corresponding RS data were divided into a training set (90%) and a testing set (10%) to calculate the $pCO_2$ reconstruction model. To ensure that the model had sufficient training samples in the coastal area, we divided the entire SCS into two regions along the 200 m depth contour. The data from these two regions were divided into training and testing sets with the same ratios listed above (9:1), which were then combined to obtain the final training and testing sets.



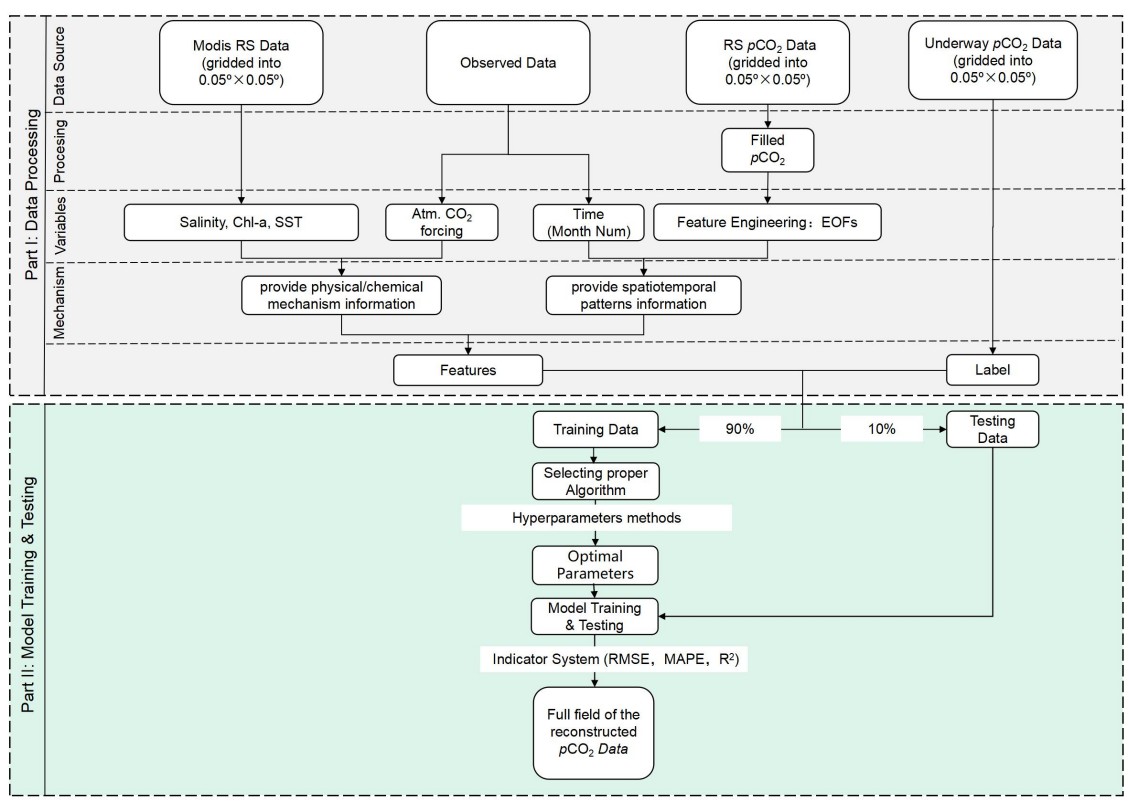

**Figure 5. Procedure for the reconstruction of surface water $p$CO$_2$ using machine learning. RS data = remote sensing data, RMSE = root mean square error, MAPE= mean absolute percentage error, and R$^2$ = coefficient of determination, and MAE = average absolute error.**

For model training and testing, we first chose a relatively reliable algorithm to undertake the $p$CO$_2$ reconstruction. After that, we determined the optimal range of the parameters using hyperparameter methods (code from https://github.com/optuna/) for the training set. The final optimal parameter values were then determined using the K-fold and cross validation method (code from https://github.com/suryanktiwari/Linear-Regression-and-K-fold-cross-validation) for the training set. These optimal parameters were applied to the chosen algorithm. Finally, the testing set was used to verify the accuracy of the $p$CO$_2$ reconstruction model produced by the training set, and some indicators of the model's accuracy were calculated. More detailed methods employed in the present study are described below.

**3.1 Remote sensing data filling**

As mentioned in the SatCO2 platform (www.SatCO2.com), the RS $p$CO$_2$ data are missing some values. Thus, we used the $p$CO$_2$ filling method suggested by Fay et al. (2021) to fill in the missing portions. First, a scaling factor for a filled month was calculated according to Eqation 1:

$$sf_{pCO2} = mean_{x,y}\left(\frac{pCO_2^{ens}}{pCO_2^{clim}}\right) \tag{1}$$



where $sf_{pCO2}$ is the scaling factor, $pCO_2^{ens}$ is the monthly RS $pCO_2$ datum, and $pCO_2^{clim}$ is the monthly climatology RS $pCO_2$
datum; $x$ and $y$ indicate that we took the area-weighted average over longitude ($x$) and latitude ($y$) to produce the monthly $sf_{pCO2}$
value. Then, the filled portion of the data can be calculated from the $pCO_2^{clim}$ data multiplied by the $sf_{pCO2}$ value (see Fay et al.
(2021) for details of this method).
Briefly, this filling method scales the climatological monthly $pCO_2$ field values to fill in the missing measurements. Therefore,
although specific values may be biased, the interpolated measurements still retain the main spatial distribution pattern of the filled
months.

**3.2 Feature engineering and selection**

As mentioned above, the $pCO_2$ filling method may bias some of the actual values. To avoid such biases, instead of directly using
the RS $pCO_2$ data as features in our reconstructed model, we used the feengineered featured data (via the EOF method) to obtain
the main spatiotemporal distribution patterns of the RS $pCO_2$ data as features in our reconstructed model. The EOF method can
reflect the spatial commonality of variables shown in the time series. For each 12 months, the cumulative variance contribution of
the first eight EOF values was consistently > 90%, indicating it that it could explain the main $pCO_2$ spatial characteristics during
each month, and we therefore selected them as features.
The feature selection in our reconstructed model can be divided into two main categories. The first one is related to the underlying
physicochemical mechanism controlling the $pCO_2$ distribution, and the other one can provide spatiotemporal information for
$pCO_2$ reconstruction. For example, the SST dominating the seasonal variation in surface water $pCO_2$ in the northern SCS (Zhai et
al., 2005; Chen et al., 2007; Li et al., 2020). Previous research (Landschützer et al., 2014; Laruelle et al., 2017; Denvil et al., 2019)
shows that Chl-$a$ plays a critical role in fitting the influence of biological activity to $pCO_2$, especially in the northern SCS
(Landschützer et al., 2014; Laruelle et al., 2017; Denvil et al., 2019). Sutton et al. (2017) suggest that the increase in atmospheric
$pCO_2$ controls the increase in seawater $pCO_2$. For the features that provide spatiotemporal information for $pCO_2$ reconstruction,
whereas in the present study we selected the first eight EOF values of $pCO_2$ as the main spatial distribution feature and monthly
information of the observed datasets as the temporal feature.

**3.3 Algorithm selection**

Ensemble learning provides one of the most powerful machine learning techniques (e.g., Zhan et al., 2022; Chen et a., 2020). It is
the process of training multiple machine learning models and combining their output to improve the reliability and accuracy of
predictions (e.g., Zhan et al., 2022; Chen et a., 2020). Different models are used as the basis to develop an optimal predictive
model. There are two main ways to employ ensemble learning: bagging (to decrease the model's variance) or boosting (to
decrease the model's bias). The random forest algorithm (code from https://scikit-learn.org/stable/) is an extension of the bagging
method as it utilizes both bagging and feature randomness to create an uncorrelated forest of decision trees. The Light Gradient
Boosting Machine (LightGBM; code from https://github.com/microsoft/LightGBM/) is a gradient boosting framework that uses





tree-based learning algorithms. LightGBM can be used for regression, classification, and other machine learning tasks; it exhibits
rapid and high-performance as a machine learning algorithm. CATBOOST (code from https://github.com/catboost/) is a gradient
boosting algorithm, which improves prediction accuracy by adjusting weights according to the data distribution and by
incorporating prior knowledge about the dataset. This can help to reduce overfitting and improve generalization performance.
From the above options, we chose three ensemble learning algorithms as the machine learning-based regression portion, and
multi-linear regression methods (Wang et al., 2021) as the linear regression portion, and we then used the K-fold and cross
validation methods to verify the applicability of the different regression algorithms in the $p\mathrm{CO_2}$ reconstruction of summer training
data in the SCS, since the greatest temporal sampling coverage occurs in summer (Table 1; Fig. 2). Results show that the
CATBOOST algorithm yields the best degree of accuracy, with an RMSE of 16 μatm; for comparison, the RMSE of LightGBM
was 27 μatm, that of Random Forest was 26 μatm, and nearly 20 μatm was found for the linear regression algorithm employed by
Wang et al. (2021). Thus, CATBOOST appears to provide a relatively reliable algorithm for $p\mathrm{CO_2}$ reconstruction.

**3.4 Evaluation metrics**
It is necessary to evaluate the accuracy of any model based on certain error metrics before applying it to specific scenarios.
Common model evaluation metrics include RMSE, MAPE , $R^2$ (coefficient of determination), and MAE.
The mean squared error (MSE) stands for the standard deviation of the residuals (prediction error), where the residuals represent
the distance between the fitted line and the data point.i.e., it stands for the degree of concentration of the reconstructed data around
the regression line. In regression analysis, RMSE is commonly used to verify experimental results. To assess bias, the RMSE
needs to combine the magnitude of the model data and is calculated as:
$$\mathrm{RMSE} = \sqrt{\frac{1}{n}\sum_{i=1}^{n}(y_i - y_{ri})^2} \quad . \tag{2}$$
where $y$ stands for the observational data, $y_r$ represents the reconstructed data, and $n$ is the number of data.
The mean absolute percentage error (MAPE) is a statistical measure used to define the accuracy of a machine learning algorithm
on a particular dataset. It is commonly used because, compared to other metrics, it uses a percentage to measure the magnitude of
the bias and is easy to understand and interpret; the lower the value of MAPE, the better a model is at forecasting. MAPE is
calculated as follows:
$$\mathrm{MAPE} = \frac{1}{n}\sum_{i=1}^{n}\frac{|y_i - y_{ri}|}{|y_i|} \tag{3}$$
The regression error metric, the coefficient of determination $R^2$, can describe the performance of a model by evaluating the
accuracy and efficiency of modeled results, i.e., it indicates the magnitude of the dependent variable calculated by the regression
model that can be explained by the independent variable, and is calculated as:



$$R^2 = 1 - \frac{\sum_{i=1}^{n}(y_i - \overline{y_i})^2}{\sum_{i=1}^{n}(y_i - y_{ri})^2} \qquad (4)$$

MAE is the average absolute difference between the field observations (true values) and model output (predicted values). The sign

of these differences is ignored so that cancellations between positive and negative values do not occur. It is calculated as:

$$MAE = \frac{1}{n}\sum_{i}^{n}|y_i - y_{ri}| \qquad (5)$$

**4 Results and discussion**

**4.1 Results**

The reconstructed $pCO_2$ fields show relatively low values in the northern coastal study region but generally shows high values in

the mid and southern basins (Fig. 6). The continuity changes of the spatiotemporal distribution can be found in the reconstruction

results (Fig. 6). The reconstructed $pCO_2$ fields show a trend of slow but sustained increase from 2003 to 2020. Spatial patterns of

$pCO_2$ change between 2003 and 2020, such that the coastal portion of the northern SCS shows relatively complex variability

because of multiple controlling factors, such as coastal upwelling, river plumes, biological activity, etc. However, $pCO_2$ values in

the mid and southern basin are relatively homogeneous, because they are mainly controlled by atmospheric $CO_2$ forcing and SST.

Temporal changes in $pCO_2$ between 2003 and 2020, are relatively large (~44 µatm) in summer and relatively small (~33 µatm) in

winter.

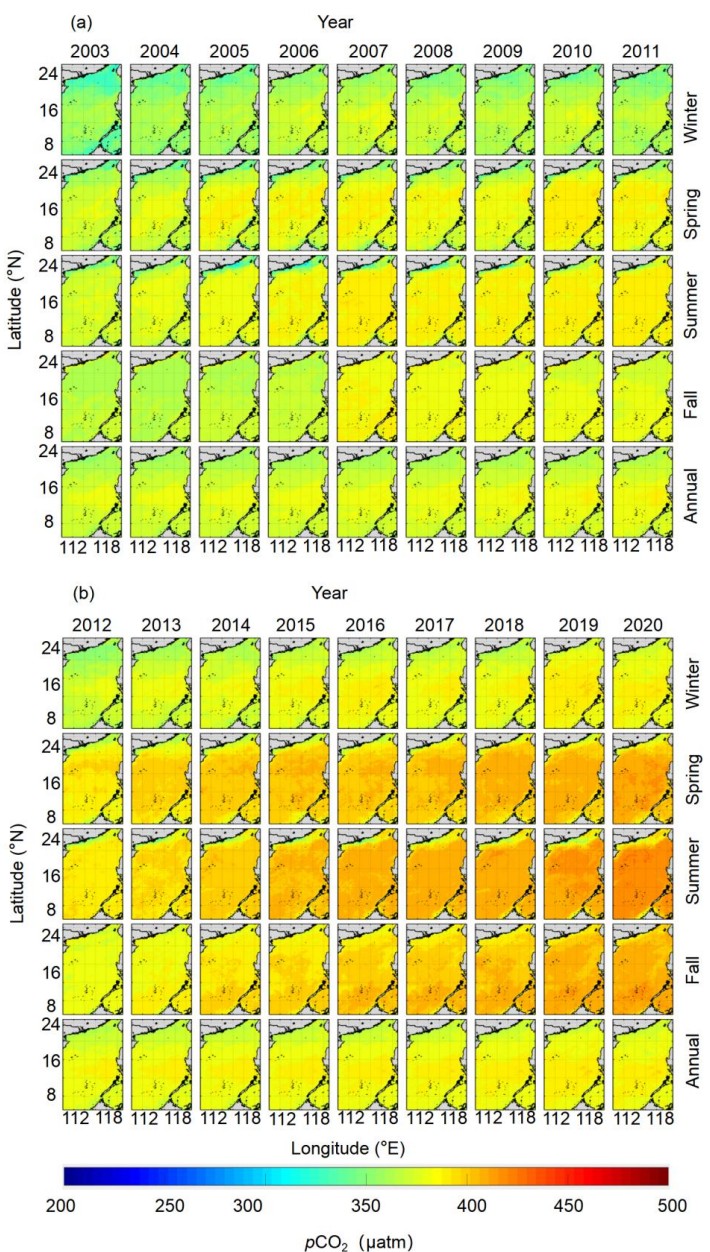

Figure 6. Reconstructed seasonal and annual pCO₂ fields in the South China Sea during the period 2003 to 2020 (a, 2003-2011; b, 2012-2020).

**4.2 Model validation**

Figure 7 compares the reconstructed and field-observed data. For the training dataset, the reconstructed $pCO_2$ fields of the four seasons fit the field-observed data well (Fig. 7), with an average RMSE of 3.43 μatm and an average MAE of ~2 μatm (Table 2).

For the testing sets, although there are some outliers, most of the reconstructed $pCO_2$ data are consistent with field-observed data,





with RMSE averaging 10.79 µatm and MAE averaging 6.30 µatm. The $R^2$ of the testing set is ca. 0.91. In terms of MAPE, the
accuracies of the four seasonal models are all around 99% (Table 2), with the highest value for spring data and the lowest value
for summer data. The greatest bias in the summer may be the influence of relatively complex regional processes, such as river
plumes and upwelling. The four evaluation metrics indicate that our reconstructed $p$CO$_2$ field is highly accurate in simulating both
the training and testing sets.

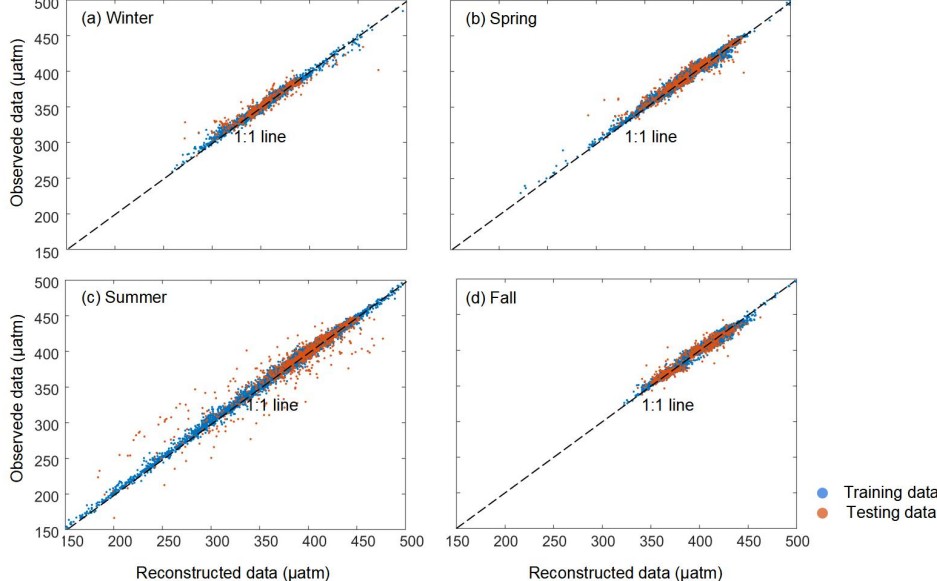


**Figure 7. Comparison between the reconstructed and the observed $p$CO$_2$ values.**
The distribution pattern of the biases between the reconstructed fields and the field observations in both training and testing
datasets can be found in Figure 8. In terms of the temporal distribution pattern, the biases are concentrated mainly in summer. For
the spatial distribution pattern, the biases in the northern coastal area are much greater than those in the basin. However, 95% of
the biases are $< \pm 10$ µatm. Therefore, our reconstruction data exhibit relatively high accuracy.



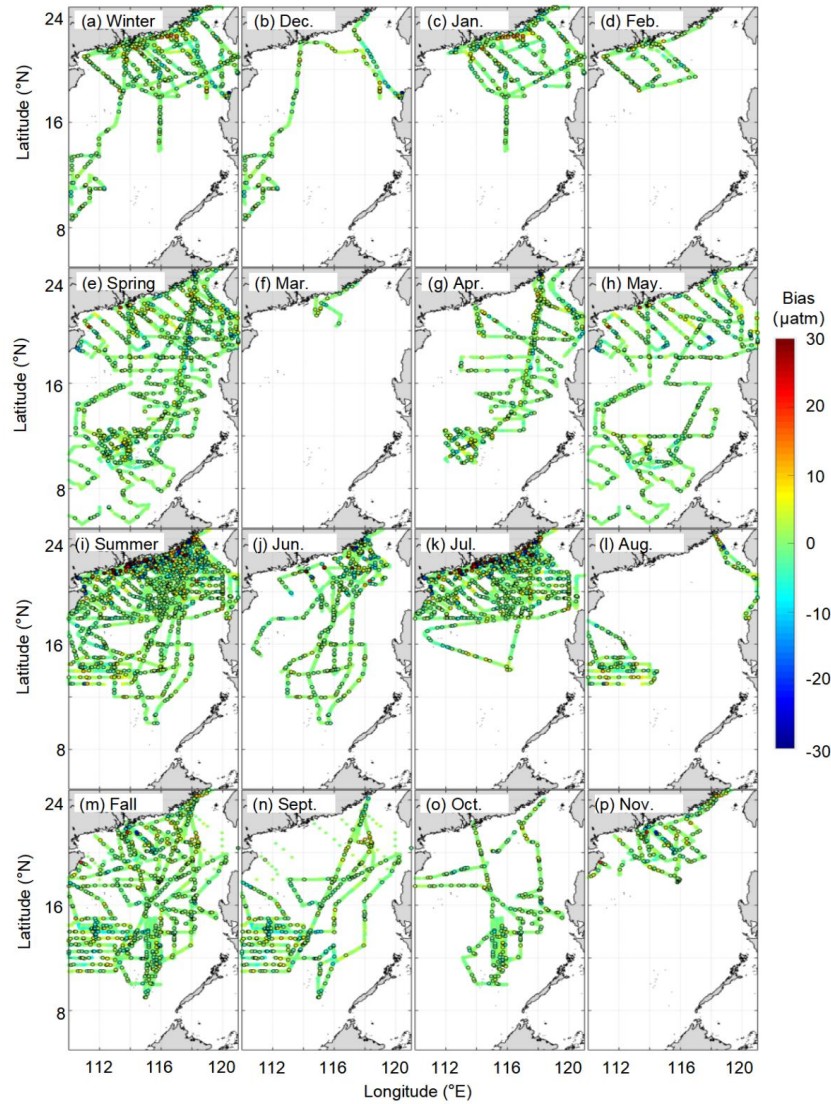


**Figure 8. Differences between the seasonal and monthly reconstructed $p$CO₂ and the observed $p$CO₂ data. The open circles**
**represent the difference between the training set and observational data, and the solid black circles represent difference**
**between the test set and observational data.**

.

Figure 9 shows the bias between our reconstructed fields and the four independent field observation datasets corresponding to the
four seasons. This validation can verify the accuracy of the reconstruction model in data months with no observations, namely the
applicability of the reconstruction model extrapolation. This comparison shows that the reconstruction model is relatively
accurate in the basin, with a near-zero bias (MAE: ~8 μatm, Fig. 9 a). The greatest bias occurs in the Pearl River plume area in
summer. The reconstruction model also has high accuracy in $tp\mathrm{CO_2}$ spatial variation trends, except in the Pearl River plume area
in summer (22–20 °N), as shown in Fig. 9 b–e). The effect of the Pearl River plume on the $p\mathrm{CO_2}$ spatial distribution in our
reconstruction model is smaller than that shown by the field-observed data. This is because at around the survey time (August
24–28, 2019), a large amount of precipitation (~30mm/day; https://psl.noaa.gov/data/gridded/data.ncep.reanalysis2.surface.html)
occurred around the Pearl River estuary region (24–20 °N), which led to intensification of the Pearl River plume, such that the
plume with relative low $\mathrm{CO_2}$ values eventually decreased the observed values. However, the monthly average runoff of the Pearl
River during that month (August, 2019; http://www.pearlwater.gov.cn/; Pearl River Plume Index in Wang et al., in preparation) is
low, indicating that our reconstruction model is still highly reliable from the a monthly average perspective. Thus, the
inconsistency between the reconstructed (monthly average) and field-observed datasets is mainly due to the differences in the time
scales of the remote sensing and the field-observed data. The reconstructed data in this study were determined on a monthly scale,
while the temporal resolution of the field-observed data was on the order of hours. It is clear that relatively pronounced short-term
changes in $p\mathrm{CO_2}$, such as the diurnal variation caused by short-term heavy precipitation, cannot be reflected in the reconstructed
data.

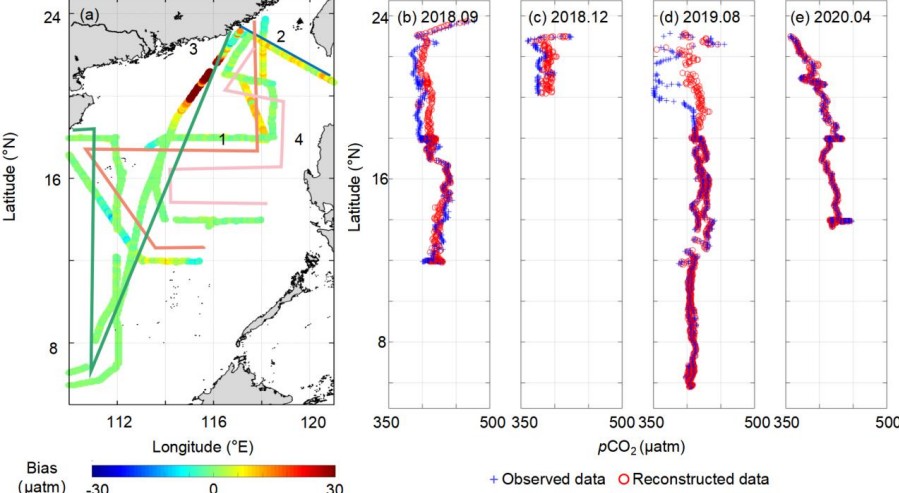


**Figure 9. Difference between the reconstructed $p\mathrm{CO_2}$ data and four independently observed datasets during the four**
**seasons. In (a), the numbers 1–4 represent September (2018.9, b), December 2018 (2018.12, c), August 2019 (2019.8, d), and**
**April 2020 (2020.4, e), respectively.**
Dai et al. (2022) produced a time-series of observed data from 2003 to 2019 at the SEATs station, which we used here to validate
the accuracy of the long-term trends of our model data (results    shown in Fig. 10). The long-term trend of reconstructed $p\mathrm{CO_2}$
data at the SEATs station are largely consistent with the observations, with differences mainly found before 2005. Thus, the
long-term trend of our reconstructed model is also highly reliable.



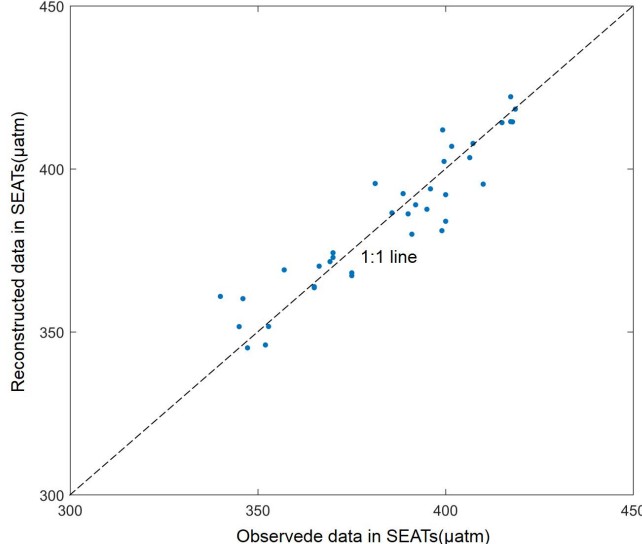


**Figure 10. Comparison of the reconstructed $p$CO$_2$ with the observations at the Southeast Asia Time Series (SEATs) station**
**(116° E, 18° N). The observed data are from Dai et al. (2022), which were calculated from dissolved inorganic carbon and**
**total alkalinity.**

**4.3 Uncertainties**
In previous studies, RMSE and MAE were mostly used to represent the uncertainties in the reconstructed data. As shown in Table
2, our reconstruction data have a high degree of accuracy, with an RMSE of ~10 μatm and MAE of ~6 μatm. However, this
expression of uncertainty ignores the sensitivity of the reconstructed model to the features; i.e., the bias that the features
themselves pass to the reconstructed model are ignored. Moreover, it is clearly unreasonable to use a single RMSE or MAE value
to represent the entire region because the spatial bias patterns clearly differ between coastal and basin areas (Figs. 8 and 9).
Thus, we here present a novel method of uncertainty calculation as shown below:
$$Uncertainty = MAX\left(\frac{\sum_{i=1,j=1}^{n} \frac{|OR\_Monthly\_Data(i,j) - Obs\_Monthly\_Data(i,j)|}{Obs\_Monthly\_Data(i,j)}}{num(i) + num(j)}\right) * 100\% * pCO2\_recon \text{ (part 1)}$$
$$+ \left(\frac{\partial pCO2}{\partial Feature}\right) dFeature \qquad \text{(part 2).} \qquad (6)$$
Equation (7) includes two parts; the first is the conservative bias between the reconstructed $p$CO$_2$ fields and the observations (part
1), and the second is the sensitivity of the reconstructed model to the features (part 2). For part 1, $OR\_Monthly\_Data(i,j)$ stands
for the monthly reconstructed data at longitude($i$) and latitude($j$), and $Obs\_Monthly\_Data(i,j)$ stands for the monthly observed
data at longitude ($i$) and latitude ($j$). Therefore, the conservative bias is the maximum value of the monthly error ratio between the
observed data and the reconstructed data. For part 2, where $dFeature$ stands for the bias of the features, we conducted a
sensitivity analysis using a chain rule to evaluate the influence of these bias in the features on $p$CO$_2$. The bias of RS $p$CO$_2$ is ~21



µatm (Table 2), the bias of SST is ~ 0.27° (Qin et al., 2014), that of SSS is ~0.33 (Wang et al., in prep.), and that of Chl-*a* is
~115% (Zhang et al., 2006). We then estimated $pCO_2$ changes due to these features' variability by constraining these features
based on our model, and computing $\frac{\partial pCO2}{\partial Feature}$. For example, for the $\frac{\partial pCO2}{\partial SST}$ part, we only changed the value of SST, and kept the
value of the other features constant, to calculate the effect of each additional unit of SST on the results of the pCO2 simulation.
These two parts were then added together to obtain the final uncertainty, and results are displayed in Figure 11. The uncertainties
are greater in the coastal area (~13 µatm), than in the basin (~10 µatm). The spatial pattern of the uncertainty is consistent with
that shown in Section 4.2.

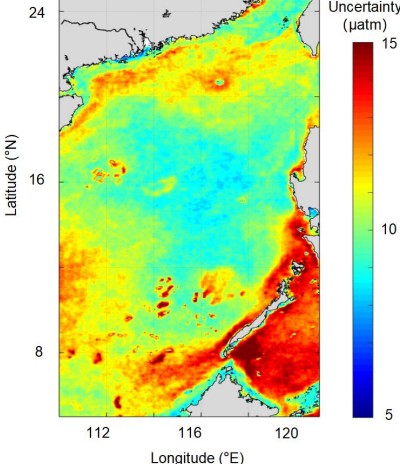


**Figure 11. Uncertainties of the reconstructed $pCO_2$ fields.**

**4.4 Spatial and temporal $pCO_2$ features**
The climatological monthly reconstructed $pCO_2$ fields are shown in Figure 12. The highest values of the reconstructed $pCO_2$ fields
occur in May and June, and the lowest value occurs in January. In winter, $pCO_2$ first decreases in December and then increases in
January; the $pCO_2$ value is ca. 325 µatm in the northern coastal area, and ca. 350 µatm in the basin. In spring, $pCO_2$ gradually
increases from the basin to the northern coastal area, and the basin high-value center gradually expands outward starting in April.
In summer, $pCO_2$ gradually declines starting in June. In fall, $pCO_2$ increases from north to south, and the southern region shows
consistently high values.

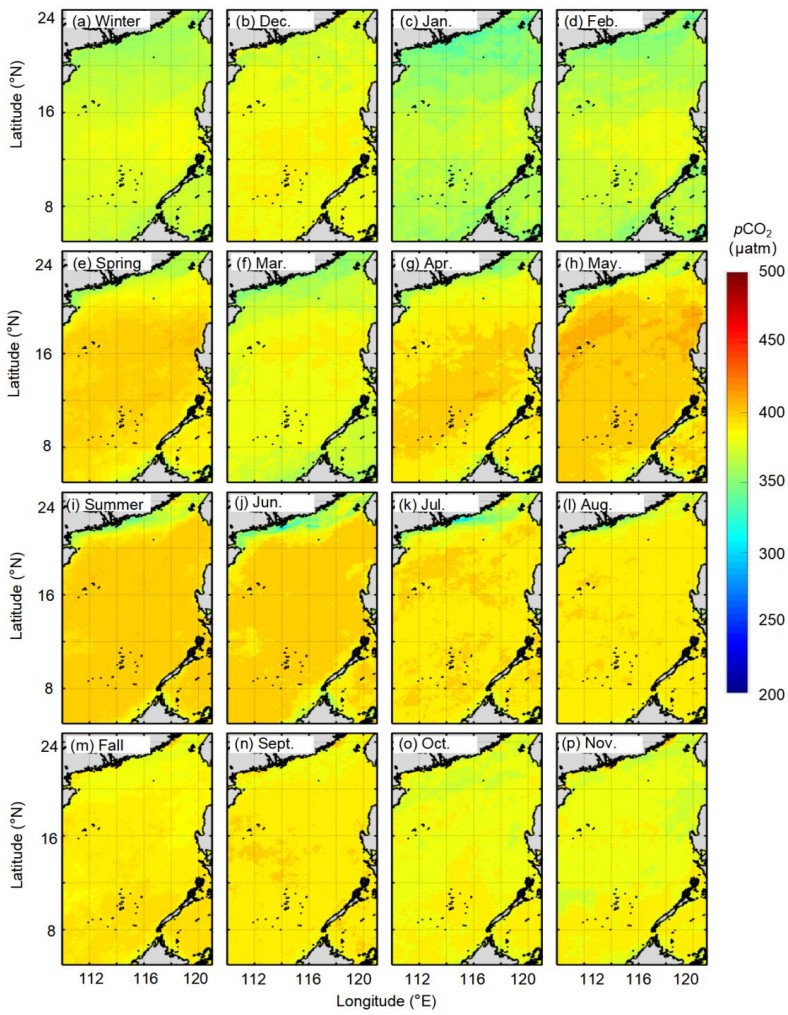

**Figure 12. Long-term (2003–2020) seasonal and monthly average $pCO_2$ field (unit: µatm).**

To better show specific regions in the northern coastal area, we zoomed in on the reconstructed $pCO_2$ fields at locations north of

18°N (Fig. 13). The reconstructed $pCO_2$ fields successfully reflect the influence of the meso-small scale processes on $pCO_2$ in this

northern coastal area of the SCS. For example, in winter, the relatively low $pCO_2$ values, which last into early spring, are mainly

controlled by the low SST, and the high $pCO_2$ around Luzon Strait affected by winter upwelling. In summer, the reconstructed

$pCO_2$ field shows that the influence of the Pearl River plume on $pCO_2$ is the strongest in July and lasts until September; it also

effectively shows the influence of coastal upwelling in the northeastern shelf (~23°N, 117°E). Thus, our reconstructed $pCO_2$ fields

clearly reflect the spatial pattern of the field observed $pCO_2$ (Fig. 3), which are generally consistent with previously reported

patterns (Li et al., 2020; Zhai et al., 2013; Gan et al., 2010).

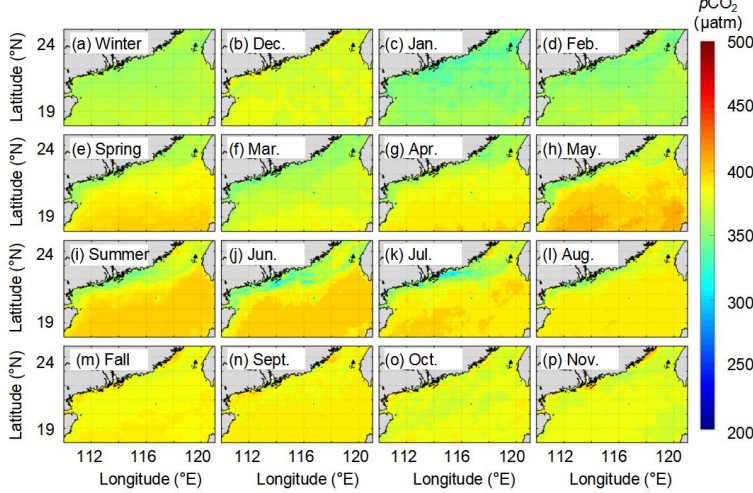

**Figure 13. Long-term (2003–2020) seasonal and monthly averaged $p$CO₂ field in the region north of 18°N (unit: μatm).**

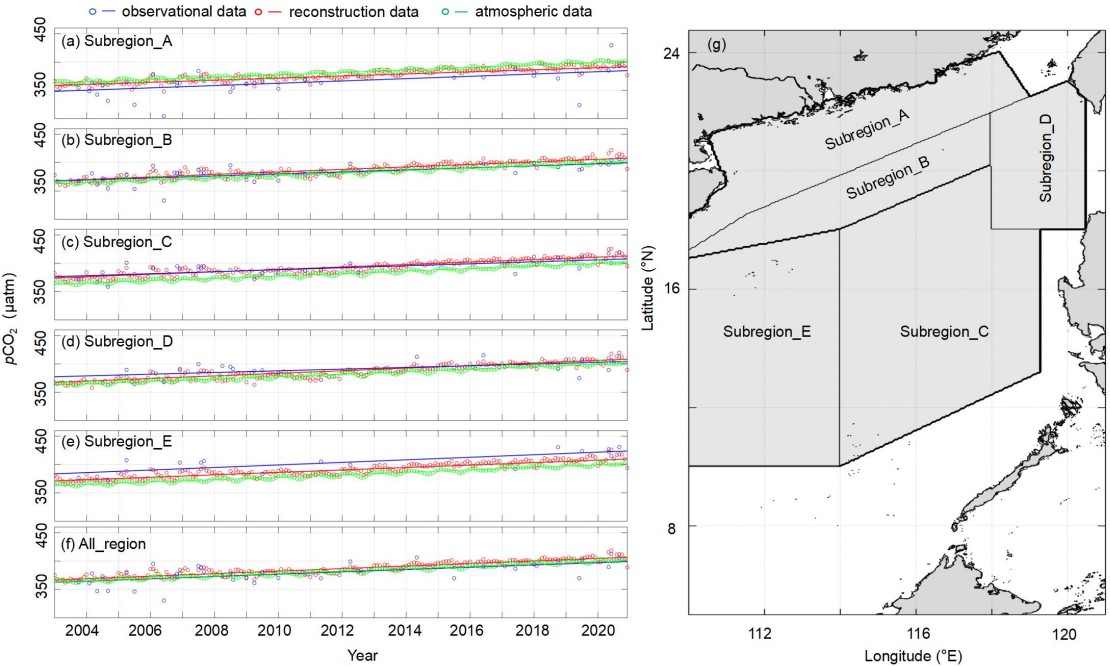

**Figure 14. Time series of Spatially averaged monthly $p$CO₂ data in five subregions (a-e) and the entire South China Sea (f) under study. The subregions are shown in (g). The lines indicate the deseasonalized long-term trend of the spatially averaged monthly $p$CO₂ data for each sub-region with the slopes shown in Table 3. The deseasonalized method can be found in Landschützer et al., 2016.**




**Table. 3 Deseasonalized long-term trend of the spatially averaged monthly $pCO_2$ data for each sub-region of the South**

**China Sea. (unit: µatm yr-1).**

|  | All_region | A_region | B_region | C_region | D_region | E_region |
|---|---|---|---|---|---|---|
| Reconstructed | 2.12±0.17 | 1.82±0.14 | 2.23±0.12 | 2.17±0.12 | 2.20±0.13 | 2.16±0.13 |
| Observation | 2.10±0.79 | 1.80±0.86 | 1.73±0.84 | 1.81±0.85 | 1.41±1.16 | 2.13±1.10 |


We divided SCS into five sub-regions according to Li et al. (2020). In Fig.14, region A stands for the northern coastal area of the
SCS, region B stands for the slope area of the northern SCS, region C stands for the SCS basin, region D stands for the region
West of the Luzon Strait, and region E stands for the slope and basin area of the western SCS. "All_region" indicates the whole
region containing the five sub-regions described above. We then calculated the deseasonalized long-term trend of spatially
averaged monthly data for each sub-region, and the results are shown in Figure 14 and Table.3. This deseasonalized trend is
consistent with that of observational data, and its uncertaintyis on the 95% confidence interval much lower than that shown by the
observational data. We can thus also infer that the long-term trend of our reconstructed data shows high reliability in all
sub-regions, and that our data can serve as an important basis for predicting future changes of $pCO_2$ in the SCS.
In Fig.14 a-e, we found that the sea surface $pCO_2$ of the entire SCS is slightly higher than the atmospheric $pCO_2$, indicating that
the SCS is a weak source of atmospheric $CO_2$. This conclusion is consistent with previous studies (e.g., Li et al., 2020). Moreover,
compared to the rate of atmospheric $CO_2$ increase (~2.2 µatm yr-1), for region A, the $pCO_2$ trend is much slower than that of
atmospheric $pCO2$, and the spatially averaged monthly mean $pCO_2$ is lower than the atmospheric $pCO2$. Thus, carbon
accumulation in this region is expected to increase in future. For regions C and E, the spatially averaged monthly mean $pCO_2$ is
higher than the atmospheric $pCO2$; thus, these two regions will still provide a weak source of atmospheric $CO_2$ in future. Finally,
whether regions B and D act as a source or sink of atmospheric $CO_2$ is influenced by seasonal changes and physical processes.

**5 Data availability**
The data (the reconstructed $CO_2$ data, the Observational $CO_2$ data, and the remote sensing derived $CO_2$ data) for this paper are
available under the link https://github.com/Elricriven/co2data (Wang et al., 2022).

**6 Conclusions**
Based on the machine learning method, we reconstructed the sea surface $pCO_2$ fields in the SCS with    high spatial resolution
(0.05*0.05º) over the last two decades (2003-2020) by calculating the statistical relationship between the underway observational
$pCO_2$ data and remote sensing data. The machine learning method used in this study was facilitated by the EOF method, because



the latter can provide spatial constraints for the data reconstruction. In addition to the typical machine learning performance
metrics, we present a novel uncertainty calculation method that incorporates the bias of both the reconstruction and the sensitivity
of reconstructed models to its features. This method effectively shows the spatiotemporal patterns of bias, and makes up for the
spatial representation of the typical performance metrics.
We validate our reconstruction with three independent testing datasets, and the results show that the bias between our
reconstruction and observational $pCO_2$ data in the SCS is relatively small (about 10 µatm). Our reconstruction successfully shows
the main features of the spatial and temporal patterns of $pCO_2$ in the SCS, indicating that we can use these reconstructed data to
further analyse the effect of meso-microscale processes (e.g., the Pearl River plume, and CCC) on sea surface $pCO_2$ in the SCS.
We divided the SCS into five sub-regions and separately calculated the deseasonalized long term trend of $pCO_2$ in each subregion,
and compared them with the long-term trend of atmospheric $pCO_2$. Our results show that the reconstructed data are consistent
with those of observational data. Moreover, the strength of the $CO_2$ sink in the northern SCS shows an increasing trend, whereas
$pCO_2$ trends in other subregions are essentially the same as that of atmospheric $pCO_2$.
This high spatiotemporal resolution of sea surface $pCO_2$ data is helpful to clarify the controlling factors of $pCO_2$ change in the
SCS and may be useful to predict changes of $CO_2$ source or sink patterns in this system.

**Author contribution**
Minhan Dai conceptualized and directed the field program of in situ observations. Xianghui Guo and Yi Xu participated in the in
situ data collection. Yan Bai provided the remote sensing-derived $pCO_2$ data. Minhan Dai, Guizhi Wang and Zhixuan Wang
developed the reconstruction method, wrote the codes, analyzed the data, and plotted the figures. Zhixuan Wang wrote the
manuscript. Minhan Dai and Xianghui Guo contributed to the writing, editing and revision of the original manuscript.

**Competing interests**
The authors declare that they have no conflict of interest.

**Acknowledgements**
We thank the support of the National Natural Science Foundation of China (grant No. 42188102, 42141001, and 41890800), and
the National Basic Research Program of China (973 Program, grant No.2015CB954000).

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
