# Peer review of "Spatial reconstruction of long-term (2003-2020) sea surface $pCO_2$ in the"

_Earth System Science Data, 2022_

## Referee Comment (RC1)

**Spatial reconstruction of long-term (2003-2020) sea surface pCO2 in the South China Sea using a machine learning based regression method aided by empirical orthogonal function analysis.**

Authors presented a machine learning approach to reconstruct ocean pCO2 over the South China Sea using the new drivers based on EOFs of Remote Sensing-derived pCO2. These new drivers contribute to the estimation accurate pCO2 product at high spatial resolution. The final product represents a monthly 0.05°x0.05° surface ocean pCO2 for the period 2003-2020. The results show a good agreement with validation data and independent observations. Authors discussed the seasonal effect on the reconstruction and mentioned seasonal processes that can affect the ocean pCO2. One of the interesting points in this work is the estimation of uncertainties. Authors introduced the estimation of uncertainties from features used in pCO2 reconstruction. The article is well structured, and it is easy to follow.

However, I found that the article missed the clarity and not all important details are presented or well explained.

Below, I listed points that need to be improved and clarified before publication.

Comments:
- The description and correct definition of data used. In your study you use the data from field survey that you call "observations" or "observed data". Also, you use remote sensing-derived data. However, it is not clear that the data from remote sensing is not direct measurements of pCO2, and it is derived product as you mentioned in 2.3 (line 156). In you abstract you speak about the comparison between "the remote sensing and observed data" (line 23) that is ambiguous. The remote sensing data are observations too and it is not exactly what was used in the paper as it was derived product. I suggest you call the data from filed survey "in situ data", and call the data derived from remote sensing "remote sensing-derived data" everywhere in the manuscript.
  Please add more details about how and what exactly was measured during the field survey. Is it the surface fugacity of CO2? If yes, you need to mention it and precise that you estimate pCO2 from fugacity.
  Please add more details on how remote sensing-derived data were produced. The website you cite in your paper www.SatCO2.com shows only homepage and it is impossible to navigate as all other webpages where we could find details about the product is forbidden. There is a little description of the product in introduction (lines 80-86), however, there is no indication that this product will be used further in the article.
  Please make corresponding changes in Figure 5: observed data to in situ data; RS pCO2 data to RS-derived pCO2 data. As you use SSS data reconstructed using ML it is incorrect to put it together with observed SST and Chl-a, or you should precise it in your figure like "ML SSS".
  Please add more information on the datasets that you introduced in lines 150-152.
- Figures' captions. Please add more information in figures' captions. Each subplot needs to be introduced in the caption.

- Tables. Please keep same number of digits in fractional part for your results in tables: Table 2, Summer RMSE has 3 digits while all other values limited by 2 digits in fractional part. Also, please use the same numbers in the text and in tables, line 163.
- Abbreviations. Please define abbreviations when you use it for the first time: for example, SSS in line 184.
- Verification of different regression algorithms. Lines 255-261. To test the capacity of different algorithm you choose the summer season due to its "greatest temporal sampling coverage". However, we can see in your article that there is a strong seasonality in pCO2 distribution. How can you be sure that algorithms will provide the same accuracy during different seasons when other features can become more important?
- Uncertainties. The method to estimate uncertainties should be presented in section 3.4 and not in the section where you discuss your results.
  In part 1 of equation 6 the function MAX does not do anything as you apply it to a scalar. What is pCO2_recon in this equation?
  Does the part 2 of equation 6 represent the sum over the features? Do you base your estimation on the error propagation method (absolute/relative error of a function)?
  It would be interesting to see the effect of individual features on pCO2 uncertainties and identify the feature that brings larger bias.
- Conclusion. Line 424, please specify which machine learning method. Line 426, please specify that you used remote sensing-derived data.
- Data pre-processing. Are the data used in ML method pre-processed: interpolated on the same grid, normalized?

- Line 148: "relatively low pCO2", what does it mean, how low is it?
- Line 164: "current algorithm", please precise, what algorithm are you talking about.
- Line 187: "our observed data", please precise which data.
- You should mention in section 2.3 that there is a section 3.1 where you explain how you fill missing points in RS-derived pCO2 product.
- Could you please provide a figure to show the distribution of training samples you mentioned in lines 201-202: "To ensure that the model had sufficient training samples in the coastal area, we divided the entire SCS into two regions along the 200 m depth contour."
- Figure 8: It is difficult to see the results for test set. The results for training set look very similar and homogeneous, I would suggest keep only test set here.
- Line 322: "The greatest bias occurs in the Pearl River plume area in summer". Could you please indicate how large is this bias?
- Line 323: what is tpCO2?
- Line 376: you say here that "the lowest value occurs in January", in the next sentence you say "pCO2 first decreases in December and then increases in January". It means that the lowest value is in December. Please clarify it.
- Line 417: "…a source or sink of atmospheric CO2 is influenced by seasonal changes and physical processes". Please specify seasonal changes and physical processes.

Typo and style:
Line 15: I would suggest using word "sparse" instead of "incomplete".

Line 37: Please change "…annually mitigates…" to "…annually mitigated…" as you refer to the concrete period of 1960-2019; or change the sentence completely.

Line 50: "Numerical ocean models of performance.." Please remove "of performance".

Line 53: I would not use the word "alternative". The data-based approaches are different methods to study ocean biogeochemistry that can be complementary to biogeochemical models.

Line 119: "CCC, yellow line in Fig. 1". There is no yellow line in Fig. 1. CCC corresponds to the green line.

Fig. 2: Please change the name of your colorbar to "number of data".

Line 145: "Spatially, the pCO2 distribution in the basin is relatively homogeneous, but shows large variability in the northern region". I suppose you meant "Spatially, the pCO2 distribution in the basin is relatively homogeneous with large variability in the northern region".

Line 288: Please change "the continuity changes.." to "the continuous changes".

Line 300: Please add that these estimations are over the seasons.

Line 322: Please change "The greatest bias" to "The largest bias".

Line 358: Please change "Equation (7)" to "Equation (6)".

Line 408: Missing space between "uncertainty" and "is".

---

## Author Comment (AC1)

Spatial reconstruction of long-term (2003-2020) sea surface $pCO_2$ in the South China Sea using a machine learning based regression method aided by empirical orthogonal function analysis.

Authors presented a machine learning approach to reconstruct ocean $pCO_2$ over the South China Sea using the new drivers based on EOFs of Remote Sensing-derived $pCO_2$. These new drivers contribute to the estimation accurate $pCO_2$ product at high spatial resolution. The final product represents a monthly 0.05°x0.05°surface ocean $pCO_2$ for the period 2003-2020. The results show a good agreement with validation data and independent observations. Authors discussed the seasonal effect on the reconstruction and mentioned seasonal processes that can affect the ocean $pCO_2$. One of the interesting points in this work is the estimation of uncertainties. Authors introduced the estimation of uncertainties from features used in $pCO_2$ reconstruction. The article is well structured, and it is easy to follow.

However, I found that the article missed the clarity and not all important details are presented or well explained. Below, I listed points that need to be improved and clarified before publication.

[Response]: We thank the reviewer for the positive comments. We have listed our point-by-point responses as of below.

Comments:

- The description and correct definition of data used. In your study you use the data from field survey that you call "observations" or "observed data". Also, you use remote sensing-derived data. However, it is not clear that the data from remote sensing is not direct measurements of $pCO_2$, and it is derived product as you mentioned in 2.3 (line 156). In you abstract you speak about the comparison between "the remote sensing and observed data" (line 23) that is ambiguous. The remote sensing data are observations too and it is not exactly what was used in the paper as it was derived product. I suggest you call the data from filed survey "in situ data", and call the data derived from remote sensing "remote sensing-derived data" everywhere in the manuscript.

[Responds]: The reviewer is right that remote sensing is also an observation tool. Revisions will be made throughout the manuscript.

- Please add more details about how and what exactly was measured during the field survey. Is it the surface fugacity of CO2? If yes, you need to mention it and precise that you estimate $pCO_2$ from fugacity.

[Responds]: We thank the reviewer for the suggestion. The details of the in situ $pCO_2$ data collections were described in Li et al. (2020). In most cruises, $pCO_2$ was measured continuously with a non-dispersive infrared spectrometer (Li-Cor® 7000) or by Cavity Ring-Down Spectroscopy (Picarro G2301) integrated in a GO-8050 system (General Oceanic Inc. USA) onboard research vessels. We will add the following information in our revision "During the cruises, sea surface $pCO_2$ was measured continuously. The measurement and data processing followed those of the SOCAT (Surface Ocean CO2 Atlas, http://www.socat.info/news.html) protocol (Li et al., 2020).".

- Please add more details on how remote sensing-derived data were produced. The website you cite in your paper www.SatCO2.com shows only homepage and it is impossible to navigate as all other webpages where we could find details about the product is forbidden. There is a little description of the product in introduction (lines 80-86), however, there is no indication that this product will be used further in the article.

[Responds]: We will add the following information to show how remote sensing (RS)-derived data were produced: "The remote sensing-derived sea surface $pCO_2$ is produced following Yu et al. (2022). The input parameters include sea surface temperature, chlorophyll-a concentration, remote sensing reflectance of three bands (Rrs412, 443, 488 nm), the temperature anomaly in longitude direction, and the theoretical thermodynamic background $pCO_2$ under corresponding SST. Although

the root mean squared errors (RMSE) associated with the RS-derived $p\text{CO}_2$ product were relatively large (21.1 µatm), it successfully showed the spatial distribution of the $p\text{CO}_2$ in China Seas (Yu et al., 2022)."

In the revision we will also add the following information "Wang et al. (2021) demonstrate that the spatial modes of remote sensing-derived data calculated using EOF are effective in providing spatial constraints on the data reconstruction and are thus adopted in this study." to explain how the RS-derived $p\text{CO}_2$ data were used in this study.

- Please make corresponding changes in Figure 5: observed data to in situ data; RS $p\text{CO}_2$ data to RS-derived $p\text{CO}_2$ data. As you use SSS data reconstructed using ML it is incorrect to put it together with observed SST and Chl-a, or you should precise it in your figure like "ML SSS".

[Responds]: Accepted and we will modify Figure 5 accordingly (Figure R1). We note that the SSS data over 2003-2020 in the South China Sea used in the present study were reconstructed based on the MODIS-Aqua remote sensing data (Wang et al., 2022). We will add this information in our revision.

[Figure]

**Figure R1. Procedure for the reconstruction of sea surface $p\text{CO}_2$ using machine learning. RS derived data = remote**

sensing-derived data, RMSE = root mean square error, MAPE= mean absolute percentage error, and $R^2$ = coefficient of determination, and MAE = average absolute error.

- Please add more information on the datasets that you introduced in lines 150-152.

[Responds]: Accepted. We will add more details as follows "In addition to the above in situ sea surface $p$CO$_2$ data, we selected four independent surveys corresponding to four seasons, September 2018 (fall), December 2018 (winter), August 2019 (summer), and April 2020 (spring), and the in situ sea surface $p$CO$_2$ data collected in these surveys are used to verify the accuracy of our reconstruction model in extrapolation to periods lacking training datasets. Furthermore, we used another dataset of sea surface $p$CO$_2$ calculated from observed dissolved inorganic carbon and total alkalinity, during 2003–2019 at the Southeast Asia Time-Series (SEATs) station (data from Dai et al., 2022) to test the long-term consistency of the reconstruction.".

- Figures' captions. Please add more information in figures' captions. Each subplot needs to be introduced in the caption.

[Responds]: Accepted. We will introduce each subplot accordingly.

- Tables. Please keep same number of digits in fractional part for your results in tables: Table 2, Summer RMSE has 3 digits while all other values limited by 2 digits in fractional part. Also, please use the same numbers in the text and in tables, line 163.

[Responds]: Accepted, and we will retain 2 significant digits after the decimal point.

- Abbreviations. Please define abbreviations when you use it for the first time: for example, SSS in line 184.

[Responds]: Accepted. SSS stands for the sea surface salinity. We will define all abbreviations at their first appearances.

- Verification of different regression algorithms. Lines 255-261. To test the capacity of different algorithm you choose the summer season due to its "greatest temporal sampling coverage". However, we can see in your article that there is a strong seasonality in $p$CO$_2$ distribution. How can you be sure that algorithms will provide the same accuracy during different seasons when other features can become more important?

[Responds]: The reviewer is correct that we performed complementary experiments for other three seasons, showing that the difference resulted from different algorithms for other seasons was minor (<2 μatm in RMSE, Table R1).

**Table R1. RMSE associated with between different algorithms in different seasons.**

| Season | Random Forest | LightGBM | CATBOOST | Multi-linear regression (Wang et al., 2021) |
|--------|---------------|----------|----------|---------------------------------------------|
| Spring | 10.65 μatm | 9.52 μatm | 8.17 μatm | NaN* |
| Summer | 26.53 μatm | 27.83 μatm | 16.15 μatm | 20.13 μatm |
| Fall | 10.34 μatm | 11.56 μatm | 10.35 μatm | NaN |
| Winter | 12.48 μatm | 12.75 μatm | 11.52 μatm | NaN |

**\*NaN stands for the missing value**

In the revision, we will add Table R1 into the MS along with the following information: "From the above options, we chose three ensemble learning algorithms as the machine learning-based regression portion, and multi-linear regression methods (Wang et al., 2021) as the linear regression portion. We then used the K-fold and cross validation methods to verify the applicability of the different regression algorithms in the $p$CO$_2$ reconstruction for seasonal training data. We show that in summer, the CATBOOST algorithm yields the best degree of accuracy, with an RMSE of 16 μatm (Table R1). For comparison, the RMSE of LightGBM was 27 μatm, and that of Random Forest was 26 μatm. The RMSE was nearly 20 μatm using the linear regression algorithm employed by Wang et al. (2021). Thus, CATBOOST appears to provide a reliable

algorithm for reconstructing $pCO_2$. Note that different algorithms for other three seasons only resulted in minor difference (~2 µatm in RMSE).".

- Uncertainties. The method to estimate uncertainties should be presented in section 3.4 and not in the section where you discuss your results. In part 1 of equation 6 the function MAX does not do anything as you apply it to a scalar. What is $pCO_2\_recon$ in this equation? Does the part 2 of equation 6 represent the sum over the features? Do you base your estimation on the error propagation method (absolute/relative error of a function)? It would be interesting to see the effect of individual features on $pCO_2$ uncertainties and identify the feature that brings larger bias.

[Responds]: Following suggestions, we will move the method to estimate uncertainties to section 3.5 and modify Equation 6 as follows (Equation R1). And Figure 11 will be modified to Figure R2 to identify the uncertainty caused by each feature.

$$Uncertainty = MAX([\frac{\sum_{i=1,j=1,k=1}^{n}\frac{|OR\_Monthly\_Data(i,j,k)-Obs\_Monthly\_Data(i,j,k)|}{Obs\_Monthly\_Data(i,j,k)}}{num(i)+num(j)}, ..., \frac{\sum_{i=1,j=1,k=n}^{n}\frac{|OR\_Monthly\_Data(i,j,k)-Obs\_Monthly\_Data(i,j,k)|}{Obs\_Monthly\_Data(i,j,k)}}{num(i)+num(j)}]) *$$

$$100\% * pCO2\_recon$$

$$+ \quad \sum_{i=1}^{n}(\frac{\partial pCO2}{\partial Feature_i})dFeature_i \qquad\qquad (R1)$$

[Figure]

**Figure R2. Uncertainties of the reconstructed sea surface $pCO_2$ fields (a, Total uncertainty in Equation 6; b. the first term of Equation 6; c. the second term of Equation 6; d stands for the $(\frac{\partial pCO2}{\partial SSS})dSSS$ in the the second term of Equation 6; e stands for the $(\frac{\partial pCO2}{\partial SST})dSST$ in the the second term of Equation 6; f stands for the $(\frac{\partial pCO2}{\partial Chl\ a})dChl\ a$ in**

the the second term of Equation 6; g stands for the $(\frac{\partial pCO2}{\partial RS\_derived\_pCO2})dRS\_derived\_pCO2$ in the the second term of Equation 6.

For the first term in Equation R1, $k$ stands for $k$th month, $OR\_Monthly\_Data(i, j, k)$ stands for the $k$th monthly reconstructed data at longitude($i$) and latitude($j$), and $Obs\_Monthly\_Data(i, j, k)$ stands for the $k$th monthly in situ data at longitude ($i$) and latitude ($j$). Therefore, the $MAX$ in first term stands for that the maximum value between the $k$ monthly bias ratios. And '$pCO_2\_recon$' stands for the reconstructed $CO_2$ data.

The second term in Equation R1 represents the sum over the features. According to Equation R1, the bias of RS derived $pCO_2$ used in the second term of Equation R1 is ~21 µatm (Table 2), the bias of SST is ~ 0.27° (Qin et al., 2014), the bias of SSS is ~0.33 (Wang et al., 2022), and the bias of Chl-a is ~115% (Zhang et al., 2006), and the results can be found in Fig. R1. of the overall uncertainty (Fig. R1 a) is greater in the coastal area (~13 µatm) than in the basin (~10 µatm). And this spatial pattern is mainly determined by the second term. The spatial distribution of the first term in Equation R1 (Fig. R1 b) calculated from a "max bias ratio" is consistent with that of $pCO_2$. The second term in Equation R1 (Fig. R1 c) is calculated from the propagation of bias of each variable. The bias of Chl $a$ (Fig. R1 f) shows the greatest effect on the reconstruction between features. Although the bias of RS derived $pCO_2$ has relatively large bias, the final influence of its bias on the reconstruction model results is negligible due to the EOF method (Fig. R1 g).

We will include this description of uncertainty in Section 4.3 of the paper.

- Conclusion. Line 424, please specify which machine learning method. Line 426, please specify that you used remote sensing-derived data.

[Responds]: Accepted. The machine learning method is CATBOOST, and the input data we used in machine learning includes remote sensing derived data (sea surface salinity, sea surface temperature, chlorophyll), the spatial patterns of $pCO_2$ calculated by Empirical Orthogonal Function, atmospheric $CO_2$, and time labels (month). We will specify these information in the revision.

- Data pre-processing. Are the data used in ML method pre-processed: interpolated on the same grid, normalized?

[Responds]: Yes, all the data used in ML were interpolated on the same grid. In the revision, we will add this information "All these data used in machine learning have been interpolated on the same grid".

- Line 148: "relatively low $pCO_2$", what does it mean, how low is it?

[Responds]: "relatively low $pCO_2$" means ~350 µatm. We will add this info in the revision.

- Line 164: "current algorithm", please precise, what algorithm are you talking about.

[Responds]: It refers to mechanic semi-analytical algorithm (MeSAA) and non-linear regression. In the revision, we will add this information.

- Line 187: "our observed data", please precise which data.

[Responds]: "our observed data" stands for the in situ data. In the revision, we will make modifications accordingly.

- You should mention in section 2.3 that there is a section 3.1 where you explain how you fill missing points in RS-derived $pCO_2$ product.

[Responds]: Accepted. We will mention this information in the end of section 2.3.

- Could you please provide a figure to show the distribution of training samples you mentioned in lines 201-202: "To ensure that the model had sufficient training samples in the coastal area, we divided the entire SCS into two regions along the 200 m depth contour."

[Responds]: Accepted, and such a figure will be added (Figure R3).

[Figure]

**Figure R3. Spatial distribution of training samples (a) and testing samples (b). The black dash line stands for the 200m depth contour.**

- Figure 8: It is difficult to see the results for test set. The results for training set look very similar and homogeneous, I would suggest keep only test set here.

[Responds]: Accepted, and we will only keep the results of test sets in Figure 8 as shown in Figure R4.

[Figure]

**Figure R4. Differences between reconstructed seasonal and monthly $pCO_2$ and the in situ $pCO_2$ for the test set (a. winter; b. December; c. January; d. February; e. Spring; f. March; g. April; h. May; i. Summer; j. June; k. July; l. August; m. Fall; n. September; o. October; p. November).**

- Line 322: "The greatest bias occurs in the Pearl River plume area in summer". Could you please indicate how large is this bias?

[Responds]: It is about 35 μatm. We will add this information in the revision.

- Line 323: what is tpCO2?

[Responds]: It should be 'the $pCO_2$', and we will remove this typo in the revision.

- Line 376: you say here that "the lowest value occurs in January", in the next sentence you say "$pCO_2$ first decreases in December and then increases in January". It means that the lowest value is in December. Please clarify it.

[Responds]: It is a typo, "$pCO_2$ decreases in December and then increases in January" should be "then increases after January". The lowest value is in January. In the revision, we will make these corrections.

- Line 417: "…a source or sink of atmospheric CO2 is influenced by seasonal changes and physical processes". Please specify seasonal changes and physical processes.

[Responds]: Accepted. We will add more details as follows "Subregion_B can be a significant sink zone of atmospheric $CO_2$ as demonstrated by its low sea surface $pCO_2$ when the Pearl River plume is spreading into a wider spatial coverage in summer. In contrast in winter when the Kuroshio intrusion is strong, both subregions B and D have high sea surface $pCO_2$, indicating that both subregions are sources of atmospheric $CO_2$."

Typo and style:

Line 15: I would suggest using word "sparse" instead of "incomplete". Line 37: Please change "…annually mitigates…" to "…annually mitigated…" as you refer to the concrete period of 1960-2019; or change the sentence completely.

[Responds]: Accepted. We will use "sparse" instead of "incomplete", and change "…annually mitigates…" to "…annually mitigated…".

Line 50: "Numerical ocean models of performance.." Please remove "of performance".

[Responds]: Accepted. We will remove "of performance" in this sentence.

Line 53: I would not use the word "alternative". The data-based approaches are different methods to study ocean biogeochemistry that can be complementary to biogeochemical models.

[Responds]: Accepted. We will rewrite this sentence as follows "data-based approaches have become an important complementary to numerical models"

Line 119: "CCC, yellow line in Fig. 1". There is no yellow line in Fig. 1. CCC corresponds to the green line.

[Responds]: The reviewer is correct, and we will make the corrections.

Fig. 2: Please change the name of your colorbar to "number of data".

[Responds]: Accepted. We will change the name of our colorbar in Fig.2 to "number of data"

Line 145: "Spatially, the $pCO_2$ distribution in the basin is relatively homogeneous, but shows large variability in the northern region". I suppose you meant "Spatially, the $pCO_2$ distribution in the basin is relatively homogeneous with large variability in the northern region".

[Responds]: Accepted. We will rewrite this sentence as your suggestion.

Line 288: Please change "the continuity changes.." to "the continuous changes".

[Responds]: Accepted. We will change "the continuity changes" to "the continuous changes".

Line 300: Please add that these estimations are over the seasons.

[Responds]: Accepted. We will add this information in the revision.

Line 322: Please change "The greatest bias" to "The largest bias".

[Responds]: Accepted. We will change "The greatest bias" to "The largest bias".

Line 358: Please change "Equation (7)" to "Equation (6)".

[Responds]: Accepted. We will make these corrections in the revision.

Line 408: Missing space between "uncertainty" and "is".

[Responds]: Accepted. We will remove this typo in the revision.

References

Wang, Z., Wang, G., Guo, X., Hu, J., and Dai, M. Reconstruction of High-Resolution Sea Surface Salinity over 2003–2020 in the South China Sea Using the Machine Learning Algorithm LightGBM Model. Remote. Sens., 14, 6147, 2022. https://doi.org/10.3390/rs14236147.

Yu, S., Song, Z., Bai, Y., and He, X.: Remote Sensing based Sea Surface partial pressure of CO2 ($pCO_2$) in China Seas (2003-2019) (2.0). Zenodo, 2022. https://doi.org/10.5281/zenodo.7372479.

---

## Author Comment (AC2)

The presented study aimed to produce monthly sea surface $p$CO$_2$ maps for the South China Sea (SCS). Given SCS is a typical temperate/subtropical marginal sea, the $p$CO$_2$ sea surface maps for this waters is necessary for understanding the CO2 flux in temperate marginal sea and even global CO2 flux. From this perspective, the study and the data it present is very meaningful. However, the manuscript still have some major flaws which do not advise me to give a yes to publishing it in its current status.

[Response]: We appreciate that the reviewer valued our study. Our point-by-point responses are listed below.

Major comments

1. The manuscript was about a dataset generation, but from the abstract and the last section, what kind of data was used as input for the method was missing.

[Responds]: Accepted, and we will add the information of input data in our revision. Note that data input includes remote sensing derived data (sea surface salinity, sea surface temperature, chlorophyll), the spatial pattern of $p$CO$_2$ calculated by Empirical Orthogonal Function, atmospheric CO$_2$, and time labels (month).

2. As I understand EOF was an important part of the method used for $p$CO$_2$ maps generation, but in the entire section of methods, no paragraph or sentence was about EOF

[Responds]: The reviewer is correct that EOF was used to obtain the main spatiotemporal pattern of the RS derived $p$CO$_2$ and then as features in our reconstructed model. Following suggestions, we will add the information of EOF as follows "The EOF reflects the spatial commonality of variables shown in the time series, which is widely used to calculate spatial patterns of climate variability (e.g. Levitus et al., 2005; Dye et al., 2020; McMonigal and Larson, 2022). Typically, the spatial commonality of variables, also named EOF modes, are found by computing the eigenvalues and eigenvectors of a spatially weighted anomaly covariance matrix of a field. Each EOF mode'?s corresponding variance represents its degree of interpretation of spatial pattern of the variable.".

3. The language of the manuscript still need some efforts. The current version contains too many redundant phrases and sentences without clear meaning and very difficult to read through and get the logical flow. Readers expect concise and precise expression in an academic paper. and there are some grammar mistake and fuzzy expression.

[Responds]: We will pay special attention on the presentation during our revisions.

4. the range of legend in nearly all the map figures were too large and cannot show the spatial gradient of $p$CO$_2$ distribution, e.g, figure 6, 8, 11, 12,13.

[Responds]: Accepted. We will adjust the range of colorbar in figures as follows (Figure R1-R7).

[Figure]

**Figure R1. Seasonal and monthly sea surface $p$CO$_2$ fields in the South China Sea. The data sources can be found in Table 1 (a. winter; b. December; c. January; d. February; e. Spring; f. March; g. April; h. May; i. Summer; j. June; k. July; l. August; m. Fall; n. September; o. October; p. November).**

[Figure]

**Figure R2. Reconstructed seasonal and annual sea surface $pCO_2$ fields in the South China Sea during the period 2003 to 2020 (a, 2003-2011; b, 2012-2020).**

[Figure]

**Figure R3. Differences between the seasonal and monthly reconstructed $p$CO$_2$ and the in situ $p$CO$_2$ data for the test set (a. winter; b. December; c. January; d. February; e. Spring; f. March; g. April; h. May; i. Summer; j. June; k. July; l. August; m. Fall; n. September; o. October; p. November). .**

[Figure]

**Figure R4. Difference between the reconstructed $p$CO$_2$ data and four independently in situ datasets during the four seasons. In (a), the numbers 1–4 represent September (2018.9, b), December 2018 (2018.12, c), August 2019 (2019.8, d), and April 2020 (2020.4, e), respectively.**

[Figure]

**Figure R5.** Uncertainties of the reconstructed $pCO_2$ fields (a, Total uncertainty in Equation 6; b. the first term of Equation 6; c. the second term of Equation 6; d stands for the $(\frac{\partial pCO2}{\partial SSS})dSSS$ in the the second term of Equation 6; e stands for the $(\frac{\partial pCO2}{\partial SST})dSST$ in the the second term of Equation 6; f stands for the $(\frac{\partial pCO2}{\partial Chl\ a})dChl\ a$ in the the second term of Equation 6; g stands for the $(\frac{\partial pCO2}{\partial RS\_derived\_pCO2})dRS\_derived\_pCO2$ in the the second term of Equation 6.

[Figure]

**Figure R6. Long-term (2003–2020) seasonal and monthly average _p_CO₂ field (unit: µatm) (a. winter; b. December; c. January; d. February; e. Spring; f. March; g. April; h. May; i. Summer; j. June; k. July; l. August; m. Fall; n. September; o. October; p. November).**

[Figure]

**Figure R7. Long-term (2003–2020) seasonal and monthly averaged $pCO_2$ field in the region north of 18°N (unit: µatm) (a. winter; b. December; c. January; d. February; e. Spring; f. March; g. April; h. May; i. Summer; j. June; k. July; l. August; m. Fall; n. September; o. October; p. November).**

5. what is the intention of including figure 4, if it is the quality of the remote sensing based $pCO_2$ maps included for further $pCO_2$ maps derivation, should the authors just need to include the information from the data distributor?

[Responds]: The reviewer is right that Figure 4 showed the quality of the RS-derived $pCO_2$ data. Following suggestions, we will remove this figure.

6. the study site section(2.1) should just serve the question "why mapping $pCO_2$ in SCS is important?", no other information is needed here.

[Responds]: We thank the reviewer for the comment. In the revision the importance of mapping $pCO_2$ in SCS will be added to section 2.1. The spatial distribution of $pCO_2$ is largely controlled by water mass missing and exchanges, thus, we retain in the introduction to the surface ocean circulation and water mass exchanges in the South China Sea in this section.

7. be consistent with the terminology, sometimes it is "in-situ", but "observational data" and "observed data" were present many times.

[Responds]: Accepted. We will unify the 'in-situ'/'observational data'/'observed data' to 'in situ data'.

8. in the abstract (line 12-14,), the importance of mapping $pCO_2$ in SCS should be addressed before presenting the method, generated data and its quality.

[Responds]: Accepted. Before presenting our method, we will add the following information "The South China Sea (SCS) is the largest marginal sea of the North Pacific Ocean, and mapping sea surface $pCO_2$ of this region is essential to better understand the spatiotemporal modes of $CO_2$ fluxes in marginal seas. In addition, we contend that the SCS is one of the most studied marginal seas in terms of carbon cycle in the world, which could thus be a model system for marginal sea carbon research" to show the importance of mapping $pCO_2$ in the SCS.

9. part of the input $pCO_2$ data of the presented study is from unpublished study (line 158), meaning not peer-reviewed.

[Responds]: We used two unpublished datasets in this paper. One of them is sea surface $pCO_2$ in China seas (0-42°N, 105-132°E) over 2003-2019 with a spatial resolution of 1 km and temporal resolution of a month (Bai et al., unpublished, line 158). This is the second version of $pCO_2$ in China seas. The first version was published on the SatCO2 website (http://www.satco2.com/index.php?m=content&c=index&a=show&catid=317&id=188) based on Bai et al. (2015). And this second version data can be cited as follows "Yu, S., Song, Z., Bai, Y., and He, X.: Remote Sensing based Sea Surface partial pressure of CO2 ($pCO_2$) in China Seas (2003-2019) (2.0). Zenodo, 2022. https://doi.org/10.5281/zenodo.7372479". Another dataset is the SSS data produced by 'Wang et al (in press)' in line 212. This paper has been accepted by Remote Sensing and its DOI number will be added in the revision as "Wang, Z., Wang, G., Guo, X., Hu, J., and Dai, M. Reconstruction of High-Resolution Sea Surface Salinity over 2003–2020 in the South China Sea Using the Machine Learning Algorithm LightGBM Model. Remote. Sens., 14, 6147, 2022. https://doi.org/10.3390/rs14236147 ".

Thus, in the revision, we will update the information accordingly.

10, line 308: Figure 7, validating the model output with the model training data gives no useful information, suggest removing this part

[Responds]: Accepted. We will only keep the results of test sets in Figure 7 as follows (Figure R8).

[Figure]

**Figure R8. Comparison between the monthly reconstructed and the in situ pCO2 values for Tesing set (monthly results were overlaid to the four seasons: (a) Winter: Dec., Jan., Feb.; (b) Spring: Mar., Apr., May; (c) Summer: Jun., Jul., Aug.; (d) Fall: Sept., Oct., Nov.).**

Minor comments

line 15-17"Using a machine learning-based method facilitated by empirical orthogonal function (EOF).... between 2003 and 2020" should specifically mention what kind of data was used for the methods input.

[Responds]: Accepted. Please refer to our response to Major Comment # 1 as of above.

linse 17- 20 "We validate our reconstruction with three independent testing datasets where,.... northern basin of the SCS." how independent are the three data set?

[Responds]: We validate our reconstruction with three independent testing datasets which are not involved model training. We will add this information in our revision.

line 22 "our reconstructions and observed data" grammar mistake.

[Responds]: Accepted. In the revision, we will rewrite this sentence as follows "The root-mean-square error (RMSE) between our reconstructed data and in situ data in TEST.1 averaged to ~10 μatm"

Line 27-28 "we present a new method to assess the uncertainty that includes the bias from the reconstruction and its sensitivity to the features,... quantifies the spatial distribution patterns of uncertainty." then the assessment method should be concisely introduced here. in addition, given this is a data presentation paper, the newly developed method should not in the highlight, unless it is a method presentation paper.

[Responds]: In the revision, we will rewrite this sentence as follows "we assess the uncertainty that includes the bias from the reconstruction and its sensitivity to the features.".

line 19 "that our reconstruction is effectively captures the main features of both the" ,check the grammar.

[Responds]: We apologize for the mistake. In the revision, we will rewrite this sentence as follows "our reconstruction effectively captures the main features of sea surface $pCO_2$ distributions in the SCS in both the spatial and temporal patterns".

line 38,, "22–26%",   I assume it should be 22%–26%.

[Responds]: The reviewer is correct, and we will make the crrection in the revision.

line 54-55:   ":The former typically use statistical interpolations and regression methods" does not fit with the neighouring sentence, rewrite it or delete it.

[Responds]: Accepted. In the revision, we will delete this sentence.

line 61- 63 ,"However, because of the complex and dynamic nature of biogeochemical and physical processes in coastal areas, characterization of sea surface $pCO_2$ and subsequently the air-sea CO2 fluxes both in time and space in marginal seas remains challenging", this sentence is too strong and undermines the motivation of presented study, rewrite it,

[Responds]: Accepted. In the revision, we will rewrite this sentence as follows "Thus, machine learning has been widely used for reconstructing sea surface $pCO_2$ for the global ocean; however, it still remains challenging to extend this method to marginal seas".

line 67: "clear need", what kind of need is clear need? a need can be strong, urgent, but not clear, need itself is a clear expression,

[Responds]: Accepted. In the revision, we will change this sentence to "Therefore, there is a strong need to achieve sea surface $pCO_2$ coverage in the SCS with a highest spatiotemporal resolution".

line 73:   "(sea surface temperature, SST; chlorophyll a, Chl a),", pay attention to journal requirements on abbreviation

[Responds]: Accepted. In the revision, we will rewrite this sentence as follows "Zhu et al. (2009) presented an empirical approach to estimate sea surface $pCO_2$ in the northern SCS in summer using satellite-derived data, including sea surface temperature (SST) and chlorophyll $a$ (Chl $a$), ...".

line 74: "underway "pay attention to the usage of underway, it is ambiguous in the manuscript.

[Responds]: Accepted. In the revision, we will change this to "in situ data" throughout the manuscript.

line 82, "the whole China Sea", where is the China Sea? do you mean all the seas in China's territory?

[Responds]: We referred to South China Sea, East China Sea, Yellow Sea, and Bohai Sea (99 - 122°E & 0 - 24°N). In the revision, we will add more details accordingly.

line 84: "(reported in Wang et al., 2021).", pay attention to the format of the reference citation

[Responds]: Accepted. We will make this correction.

line 84: "Bai et al. (unpublished) subsequently", if the work is not publised, then it should not be cited or discussed, as it is not peer-reviewed.

[Responds]: This dataset is an updated version based on Bai et al. (2015). Please refer to our response to Major Comment # 9 as of above. In the revision, we will update this citation to "Yu et al. (2022)".

line 94-96: include the input data here.

[Responds]: Accepted. In the revision, we will add some details of input data as follows "and selecting the remote sensing derived data (sea surface salinity, sea surface temperature, chlorophyll), the spatial patterns of $pCO_2$ calculated by Empirical Orthogonal Function, atmospheric $CO_2$, and time labels (month) as input data".

line 137-138 : there is no asterisk in the table and the meaning of the asterisk led note is not clear.

[Responds]: Accepted. In the revision, we will modify this Table as follows (Table R1).

**Table R1. Summary of the seasonal in situ data of sea surface $pCO_2$ in the South China Sea for the period 2003-2020 used in this study.**

| Season | Spring | | | | Summer | |
|---|---|---|---|---|---|---|
| | March | April | May | June | July | August |
| Cruise time | 2004.03 | 2005.04 2008.04 2009.04 2012.04 2020.04* | 2004.05 2011.05 2014.05 2020.05* | 2006.06 2016.06 2017.06* 2019.06* 2020.06* | 2004.07 2005.07 2007.07 2008.07 2009.07 2012.07 2015.07* 2019.07* | 2007.08 2008.08 2019.08* |

| Season | Fall | | | Winter | | |
|---|---|---|---|---|---|---|
| | September | October | November | December | January | February |
| Cruise time | 2004.09 2007.09 2008.09 2020.09* | 2003.10 2006.10 | 2006.11 2010.11 | 2006.12 | 2009.01 2010.01 2018.01 | 2004.02 2006.02 |
| Data source | Li et al. (2020) *This study | | | | | |

line 144 "Figure 3 shows the spatial and temporal distributions of surface water $pCO_2$.", the spatial distribution of in-situ measurements or data from other source?

[Responds]: Figure 3 shows the spatial distribution of in-situ measurements. In the revision, we will add more details as follows: "Figure 3 shows the spatial and temporal distributions of sea surface water $pCO_2$ of in situ measurements." .

line 157:  how the remote sensing-derived $pCO_2$ data were derived? which methods, what is the quality? and output from unpublished study should not be used.

[Responds]: This dataset is an updated version based on Bai et al. (2015). Please refer to our response to Major Comment # 9 as of above. In the revision, we will change this citation to "Yu et al. (2022)", and will add more details of this dataset as follows "Yu et al. (2022) subsequently used a non-linear regression to develop a retrieval algorithm for seawater $pCO_2$ in the China Sea, and the satellite-derived $pCO_2$ data from 2003-2018 were provided by the SatCO2 platform (www.SatCO2.com). In the retrieval algorithm of Yu et al. (2022), the input parameters are sea surface temperature, chlorophyll-a concentration, remote sensing reflectance of three bands (Rrs412, 443, 488 nm), the temperature anomaly in longitude direction, and the theoretical thermodynamic background $pCO_2$ under corresponding SST. Although the root mean squared errors (RMSE) for the $pCO_2$ were relatively large (21.1 μatm), it successfully showed the spatial distribution of the $pCO_2$ in China Seas (Yu et al., 2022)." .

line 184-187: "Wang et al. (in preparation) found a relatively high differential between the....observed data", meaning of this super long sentence is not clear.

[Responds]: Accepted. In the revision, we will modify this sentence as follows "For the sea surface salinity (SSS) data, Wang et al. (2022) found relatively large difference between the different open source SSS databases (i.e., multi-satellite fusion data from https://podaac.jpl.nasa.gov/; model data from https://climatedataguide.ucar.edu/; multidimensional covariance model data from https://resources.marine.copernicus.eu/) and the in situ SSS data." .

line 198 "$pCO_2$ filling method of", should explain the filling method here!

[Responds]: Accepted. In the revision, we will modify this sentence as follows "Secondly, we used the $pCO_2$ filling method according to Fay et al. (2021) to fill the missing $pCO_2$ measurements with the RS $pCO_2$ data, and this filling method can be found in section 3.1." because that the $pCO_2$ filling method would be explained in section 3.1.

line 201: "$pCO_2$ reconstruction model"   $pCO_2$ reconstruction was used many times in the manuscript, but sea surface $pCO_2$ is not something one can reconstruct, it is a properties or variable of of the sea water, one can measure it ,describe it, retrieve its distribution, but not reconstruct $pCO_2$ itself.   So, please pay attention to the verb usage.

[Responds]: Accepted. In the revision, we will change "$pCO_2$ reconstruction model" to "$pCO_2$ retrieve algorithm"

References

Dye, A. W., Rastogi, B., Clemesha, R. E. S., Kim, J. B., Samelson, R. M., Still, C. J., & Williams, A. P.: Spatial patterns and trends of summertime low cloudiness for the Pacific Northwest, 1996–2017. Geophysical Research Letters, 47, e2020GL088121, 2020. https://doi.org/10.1029/2020GL088121

Levitus, S., Antonov, J. I., Boyer, T. P., Garcia, H. E., and Locarnini, R. A.: EOF analysis of upper ocean heat content, 1956–2003, Geophys. Res. Lett., 32, L18607, 2005. doi:10.1029/2005GL023606.

McMonigal, K., & Larson, S. M.: ENSO explains the link between Indian Ocean dipole and Meridional Ocean heat transport. Geophysical Research Letters, 49, e2021GL095796, 2022. https://doi.org/10.1029/2021GL095796

Wang, Z., Wang, G., Guo, X., Hu, J., and Dai, M. Reconstruction of High-Resolution Sea Surface Salinity over 2003–2020 in the South China Sea Using the Machine Learning Algorithm LightGBM Model. Remote. Sens., 14, 6147, 2022. https://doi.org/10.3390/rs14236147.

Yu, S., Song, Z., Bai, Y., and He, X.: Remote Sensing based Sea Surface partial pressure of CO2 (pCO2) in China Seas (2003-2019) (2.0). Zenodo, 2022. https://doi.org/10.5281/zenodo.7372479.

---

## Author Response (AR1)

**[Response to Reviewer#1]**

Spatial reconstruction of long-term (2003-2020) sea surface $p\text{CO}_2$ in the South China Sea using a machine learning based regression method aided by empirical orthogonal function analysis.

Authors presented a machine learning approach to reconstruct ocean $p\text{CO}_2$ over the South China Sea using the new drivers based on EOFs of Remote Sensing-derived $p\text{CO}_2$. These new drivers contribute to the estimation accurate $p\text{CO}_2$ product at high spatial resolution. The final product represents a monthly 0.05°x0.05°surface ocean $p\text{CO}_2$ for the period 2003-2020. The results show a good agreement with validation data and independent observations. Authors discussed the seasonal effect on the reconstruction and mentioned seasonal processes that can affect the ocean $p\text{CO}_2$. One of the interesting points in this work is the estimation of uncertainties. Authors introduced the estimation of uncertainties from features used in $p\text{CO}_2$ reconstruction. The article is well structured, and it is easy to follow.

However, I found that the article missed the clarity and not all important details are presented or well explained. Below, I listed points that need to be improved and clarified before publication.

[Response]: We thank the reviewer for the positive comments. We have listed our point-by-point responses as of below.

Comments:

- The description and correct definition of data used. In your study you use the data from field survey that you call "observations" or "observed data". Also, you use remote sensing-derived data. However, it is not clear that the data from remote sensing is not direct measurements of $p\text{CO}_2$, and it is derived product as you mentioned in 2.3 (line 156). In you abstract you speak about the comparison between "the remote sensing and observed data" (line 23) that is ambiguous. The remote sensing data are observations too and it is not exactly what was used in the paper as it was derived product. I suggest you call the data from filed survey "in situ data", and call the data derived from remote sensing "remote sensing-derived data" everywhere in the manuscript.

[Responds]: The reviewer is right that remote sensing is also an observation tool. Revisions have been made throughout the manuscript.

- Please add more details about how and what exactly was measured during the field survey. Is it the surface fugacity of CO2? If yes, you need to mention it and precise that you estimate $p\text{CO}_2$ from fugacity.

[Responds]: We thank the reviewer for the suggestion. The details of the in situ $p\text{CO}_2$ data collections were described in Li et al. (2020). In most cruises, $p\text{CO}_2$ was measured continuously with a non-dispersive infrared spectrometer (Li-Cor® 7000) or by Cavity Ring-Down Spectroscopy (Picarro G2301) integrated in a GO-8050 system (General Oceanic Inc. USA) onboard research vessels. We have added the following information in our revision "During the cruises, sea surface $p\text{CO}_2$ was measured underway. The measurement and data processing followed the SOCAT (Surface Ocean CO2 Atlas) protocol (Li et al., 2020)." (Lines 140-141).

- Please add more details on how remote sensing-derived data were produced. The website you cite in your paper www.SatCO2.com shows only homepage and it is impossible to navigate as all other webpages where we could find details about the product is forbidden. There is a little description of the product in introduction (lines 80-86), however, there is no indication that this product will be used further in the article.

[Responds]: We have added the following information to show how remote sensing (RS)-derived data were produced: "Yu et al. (2022) subsequently used a non-linear regression method to develop a retrieval algorithm for seawater $p\text{CO}_2$ in the China Seas, and the RS-derived $p\text{CO}_2$ data from 2003-2018 were provided by the SatCO2 platform (www.SatCO2.com). In the retrieval algorithm of Yu et al. (2022), the input parameters include sea surface temperature,

chlorophyll-a concentration, remote sensing reflectance of three bands (Rrs412, 443, 488 nm), the temperature anomaly in the longitude direction, and the theoretical thermodynamic background $pCO_2$ under corresponding SST. Although the RMSE associated with the RS-derived $pCO_2$ product was relatively large (21.1 µatm), it successfully showed major spatial patterns of the sea surface $pCO_2$ in the China Seas (Yu et al., 2022)." (Lines 84-90).

In the revision we have also added the following information "Wang et al. (2021) demonstrate that the spatial modes of RS-derived data calculated using EOF are effective in providing spatial constraints on the data reconstruction and are thus adopted in this study." in lines 93-95 to explain how the RS-derived $pCO_2$ data were used in this study.

- Please make corresponding changes in Figure 5: observed data to in situ data; RS $pCO_2$ data to RS-derived $pCO_2$ data. As you use SSS data reconstructed using ML it is incorrect to put it together with observed SST and Chl-a, or you should precise it in your figure like "ML SSS".

[Responds]: Accepted and we have modified Figure 5 accordingly (Figure R1). We note that the SSS data over 2003-2020 in the South China Sea used in the present study were reconstructed based on the MODIS-Aqua remote sensing data (Wang et al., 2022). We have added this information in our revision in Lines 197-199.

[Figure]

**Figure R1. Procedure for the reconstruction of sea surface $pCO_2$ using machine learning. RS derived data = remote sensing-derived data, RMSE = root mean square error, MAPE= mean absolute percentage error, and $R^2$ = coefficient of determination, and MAE = average absolute error.**

- Please add more information on the datasets that you introduced in lines 150-152.

[Responds]: Accepted. We have added more details as follows "In addition to the above in situ sea surface $pCO_2$ data, to verify the accuracy of our reconstruction model in extrapolation to periods lacking training datasets, we selected the in situ sea surface $pCO_2$ data collected in four independent surveys corresponding to four seasons, September 2018 (fall), December 2018 (winter), August 2019 (summer), and April 2020 (spring). Furthermore, we used another dataset of sea surface $pCO_2$ calculated from observed dissolved inorganic carbon and total alkalinity during 2003–2019 at the Southeast Asia Time-Series (SEAts) station (data from Dai et al., 2022) to test the long-term consistency of the reconstruction." in Lines 160-165.

- Figures' captions. Please add more information in figures' captions. Each subplot needs to be introduced in the caption.

[Responds]: Accepted. We have introduced each subplot in the revised manuscript accordingly.

- Tables. Please keep same number of digits in fractional part for your results in tables: Table 2, Summer RMSE has 3 digits while all other values limited by 2 digits in fractional part. Also, please use the same numbers in the text and in tables, line 163.

[Responds]: Accepted, and we have retained 2 significant digits after the decimal point in the revised manuscript.

- Abbreviations. Please define abbreviations when you use it for the first time: for example, SSS in line 184.

[Responds]: Accepted. SSS stands for the sea surface salinity. In the revision, we have defined all abbreviations at their first appearances.

- Verification of different regression algorithms. Lines 255-261. To test the capacity of different algorithm you choose the summer season due to its "greatest temporal sampling coverage". However, we can see in your article that there is a strong seasonality in $pCO_2$ distribution. How can you be sure that algorithms will provide the same accuracy during different seasons when other features can become more important?

[Responds]: The reviewer is correct that we performed complementary experiments for other three seasons, showing that the difference resulted from different algorithms for other seasons was minor (<2 μatm in RMSE, Table R1).

**Table R1. RMSE associated with between different algorithms in different seasons.**

| Season | Random Forest | LightGBM | CATBOOST | Multi-linear regression (Wang et al., 2021) |
|--------|---------------|----------|----------|---------------------------------------------|
| Spring | 10.65 μatm | 9.52 μatm | 8.17 μatm | NaN* |
| Summer | 26.53 μatm | 27.83 μatm | 16.15 μatm | 20.13 μatm |
| Fall | 10.34 μatm | 11.56 μatm | 10.35 μatm | NaN |
| Winter | 12.48 μatm | 12.75 μatm | 11.52 μatm | NaN |

**\*NaN stands for the missing value**

In the revision, we have added Table R1 into the MS along with the following information: "From the above options, we chose three ensemble learning algorithms as the machine learning-based regression portion, and multi-linear regression methods (Wang et al., 2021) as the linear regression portion. We then used the K-fold and cross validation methods to verify the applicability of different regression algorithms in the $pCO_2$ reconstruction for seasonal training data. The results show that in summer the CATBOOST algorithm yields the best degree of accuracy with an RMSE of 16 μatm (Table R1). In contrast, the RMSE of LightGBM was 27 μatm, and that of Random Forest was 26 μatm. The RMSE was nearly 20 μatm using the linear regression algorithm employed by Wang et al. (2021). Thus, CATBOOST appears to provide a reliable algorithm for reconstructing $pCO_2$. In other three seasons, however, different algorithms resulted in minor differences (~2 μatm in RMSE)." in Lines 273-279.

- Uncertainties. The method to estimate uncertainties should be presented in section 3.4 and not in the section where you discuss your results. In part 1 of equation 6 the function MAX does not do anything as you apply it to a scalar. What is $pCO_2\_recon$ in this equation? Does the part 2 of equation 6 represent the sum over the features? Do you base your estimation on the error propagation method (absolute/relative error of a function)? It would be interesting to see the effect of individual features on $pCO_2$ uncertainties and identify the feature that brings larger bias.

[Responds]: Following suggestions, we have moved the method to estimate uncertainties to section 3.5 and modify Equation 6 as follows (Equation R1) in the revision. And Figure 11 have been modified to Figure R2 to identify the uncertainty caused by each feature in the revision.

$$Uncertainty = MAX([\frac{\sum_{i=1,j=1,k=1}^{n}\frac{|OR\_Monthly\_Data(i,j,k)-Obs\_Monthly\_Data(i,j,k)|}{Obs\_Monthly\_Data(i,j,k)}}{num(i)+num(j)}, ..., \frac{\sum_{i=1,j=1,k=n}^{n}\frac{|OR\_Monthly\_Data(i,j,k)-Obs\_Monthly\_Data(i,j,k)|}{Obs\_Monthly\_Data(i,j,k)}}{num(i)+num(j)}]) *$$

$$100\% * pCO2\_recon$$

$$+ \qquad \sum_{i=1}^{n}(\frac{\partial pCO2}{\partial Feature_i})dFeature_i \qquad\qquad . \qquad\qquad (R1)$$

[Figure]

**Figure R2. Uncertainties of the reconstructed sea surface $pCO_2$ fields (a, Total uncertainty in Equation 6; b. the first term of Equation 6; c. the second term of Equation 6; d stands for the $(\frac{\partial pCO2}{\partial SSS})dSSS$ in the the second term of Equation 6; e stands for the $(\frac{\partial pCO2}{\partial SST})dSST$ in the the second term of Equation 6; f stands for the $(\frac{\partial pCO2}{\partial Chl\,a})dChl\,a$ in the the second term of Equation 6; g stands for the $(\frac{\partial pCO2}{\partial RS\_derived\_pCO2})dRS\_derived\_pCO2$ in the the second term of Equation 6.**

For the first term in Equation R1, $k$ stands for $k$th month, $OR\_Monthly\_Data(i,j,k)$ stands for the $k$th monthly reconstructed data at longitude($i$) and latitude($j$), and $Obs\_Monthly\_Data(i,j,k)$ stands for the $k$th monthly in situ data at longitude ($i$) and latitude ($j$). Therefore, the $MAX$ in first term stands for that the maximum value between the $k$ monthly bias ratios. And '$pCO_2\_recon$' stands for the reconstructed $CO_2$ data.

The second term in Equation R1 represents the sum over the features. According to Equation R1, the bias of RS derived $pCO_2$ used in the second term of Equation R1 is ~21 μatm (Table 2), the bias of SST is ~ 0.27° (Qin et al., 2014), the bias of SSS is ~0.33 (Wang et al., 2022), and the bias of Chl-a is ~115% (Zhang et al., 2006), and the results can be found in Fig. R1. of the overall uncertainty (Fig. R1 a) is greater in the coastal area (~13 μatm) than in the basin (~10 μatm). And this spatial pattern is mainly determined by the second term. The spatial distribution of the first term in Equation R1 (Fig. R1 b) calculated from a "max bias ratio" is consistent with that of $pCO_2$. The second term in Equation R1 (Fig. R1 c) is calculated from the propagation of bias of each variable. The bias of Chl $a$ (Fig. R1 f) shows the greatest effect on the reconstruction between features. Although the bias of RS derived $pCO_2$ has relatively large bias, the final influence of its bias on the reconstruction model results is negligible due to the EOF method (Fig. R1 g).
We have included this description of uncertainty in Section 4.3 of the revision.

- Conclusion. Line 424, please specify which machine learning method. Line 426, please specify that you used remote sensing-derived data.
[Responds]: Accepted. The machine learning method is CATBOOST, and the input data we used in machine learning includes remote sensing derived data (sea surface salinity, sea surface temperature, chlorophyll), the spatial patterns of $pCO_2$ calculated by Empirical Orthogonal Function, atmospheric $CO_2$, and time labels (month). We have specified these information in the revision.

- Data pre-processing. Are the data used in ML method pre-processed: interpolated on the same grid, normalized?
[Responds]: Yes, all the data used in ML were interpolated on the same grid. In the revision, we have added this information "Note that all these data used in machine learning have been interpolated on the same grid." in Line 214.

- Line 148: "relatively low $pCO_2$", what does it mean, how low is it?
[Responds]: "relatively low $pCO_2$" means ~350 μatm. We have added this info in the revision (Line 157).

- Line 164: "current algorithm", please precise, what algorithm are you talking about.
[Responds]: It refers to mechanic semi-analytical algorithm (MeSAA) and non-linear regression. In the revision, we have added this information (Line 178) .

- Line 187: "our observed data", please precise which data.
[Responds]: "our observed data" stands for the in situ data. In the revision, we have made modifications accordingly.

- You should mention in section 2.3 that there is a section 3.1 where you explain how you fill missing points in RS-derived $pCO_2$ product.
[Responds]: Accepted. We have mentioned this information in the end of section 2.3.

- Could you please provide a figure to show the distribution of training samples you mentioned in lines 201-202: "To ensure that the model had sufficient training samples in the coastal area, we divided the entire SCS into two regions along the 200 m depth contour."
[Responds]: Accepted, and such a figure (Figure R3) have been added in the revision as Figure 5.

[Figure]

**Figure R3. Spatial distribution of training samples (a) and testing samples (b). The black dash line stands for the 200m depth contour.**

- Figure 8: It is difficult to see the results for test set. The results for training set look very similar and homogeneous, I would suggest keep only test set here.

[Responds]: Accepted, and we have only kept the results of test sets in Figure 8 as shown in Figure R4.

[Figure]

**Figure R4. Differences between reconstructed seasonal and monthly $pCO_2$ and the in situ $pCO_2$ for the test set (a. winter; b. December; c. January; d. February; e. Spring; f. March; g. April; h. May; i. Summer; j. June; k. July; l. August; m. Fall; n. September; o. October; p. November).**

- Line 322: "The greatest bias occurs in the Pearl River plume area in summer". Could you please indicate how large is this bias?

[Responds]: It is about 35 µatm. We have added this information in the revision (Line 359).

- Line 323: what is tpCO2?

[Responds]: It should be 'the $pCO_2$', and we have removed this typo in the revision.

- Line 376: you say here that "the lowest value occurs in January", in the next sentence you say "$pCO_2$ first decreases in December and then increases in January". It means that the lowest value is in December. Please clarify it.

[Responds]: It is a typo, "$pCO_2$ decreases in December and then increases in January" should be "then increases after January". The lowest value is in January. In the revision, we have made these corrections.

- Line 417: "...a source or sink of atmospheric CO2 is influenced by seasonal changes and physical processes". Please specify seasonal changes and physical processes.

[Responds]: Accepted. We have added more details as follows "Subregion_B can be a zone of significant sink of atmospheric $CO_2$ as demonstrated by its low sea surface $pCO_2$ when the Pearl River plume spreads more widely in summer. In contrast, in winter when the Kuroshio intrusion is strong, both Subregions B and D have high sea surface $pCO_2$, indicating both subregions are sources of atmospheric $CO_2$." in the revision (Lines 452-455).

Typo and style:

Line 15: I would suggest using word "sparse" instead of "incomplete". Line 37: Please change "...annually mitigates..." to "...annually mitigated..." as you refer to the concrete period of 1960-2019; or change the sentence completely.

[Responds]: Accepted. We have used "sparse" instead of "incomplete", and change "...annually mitigates..." to "...annually mitigated..." (Line 15).

Line 50: "Numerical ocean models of performance.." Please remove "of performance".

[Responds]: Accepted. We have removed "of performance" in this sentence (Line 50).

Line 53: I would not use the word "alternative". The data-based approaches are different methods to study ocean biogeochemistry that can be complementary to biogeochemical models.

[Responds]: Accepted. We have rewritten this sentence as follows "data-based approaches have become an important complementary to numerical models" (Lines 55-56).

Line 119: "CCC, yellow line in Fig. 1". There is no yellow line in Fig. 1. CCC corresponds to the green line.

[Responds]: The reviewer is correct, and we have made the corrections.

Fig. 2: Please change the name of your colorbar to "number of data".

[Responds]: Accepted. We have change the name of our colorbar in Fig.2 to "number of data"

Line 145: "Spatially, the $pCO_2$ distribution in the basin is relatively homogeneous, but shows large variability in the northern region". I suppose you meant "Spatially, the $pCO_2$ distribution in the basin is relatively homogeneous with large variability in the northern region".

[Responds]: Accepted. We have rewritten this sentence as your suggestion (Lines 154-155).

Line 288: Please change "the continuity changes.." to "the continuous changes".

[Responds]: Accepted. We have changed "the continuity changes" to "the continuous changes" (Line 324).

Line 300: Please add that these estimations are over the seasons.

[Responds]: Accepted. We have added this information in the revision.

Line 322: Please change "The greatest bias" to "The largest bias".

[Responds]: Accepted. We have changed "The greatest bias" to "The largest bias" (Line 359).

Line 358: Please change "Equation (7)" to "Equation (6)".

[Responds]: Accepted. We have made these corrections in the revision.

Line 408: Missing space between "uncertainty" and "is".

[Responds]: Accepted. We have removed this typo in the revision (Line 392).

References

Wang, Z., Wang, G., Guo, X., Hu, J., and Dai, M. Reconstruction of High-Resolution Sea Surface Salinity over 2003–2020 in the South China Sea Using the Machine Learning Algorithm LightGBM Model. Remote. Sens., 14, 6147, 2022. https://doi.org/10.3390/rs14236147.

Yu, S., Song, Z., Bai, Y., and He, X.: Remote Sensing based Sea Surface partial pressure of CO2 ($pCO_2$) in China Seas (2003-2019) (2.0). Zenodo, 2022. https://doi.org/10.5281/zenodo.7372479.

**[Response to Reviewer#2]**

The presented study aimed to produce monthly sea surface $pCO_2$ maps for the South China Sea (SCS). Given SCS is a typical temperate/subtropical marginal sea, the $pCO_2$ sea surface maps for this waters is necessary for understanding the CO2 flux in temperate marginal sea and even global CO2 flux. From this perspective, the study and the data it present is very meaningful. However, the manuscript still have some major flaws which do not advise me to give a yes to publishing it in its current status.

[Response]: We appreciate that the reviewer valued our study. Our point-by-point responses are listed below.

Major comments

1. The manuscript was about a dataset generation, but from the abstract and the last section, what kind of data was used as input for the method was missing.

[Responds]: Accepted, and we have added the information of input data in the abstract and the last section of revision. Note that data input includes remote sensing derived data (sea surface salinity, sea surface temperature, chlorophyll), the spatial pattern of $pCO_2$ calculated by Empirical Orthogonal Function, atmospheric $CO_2$, and time labels (month).

2. As I understand EOF was an important part of the method used for $pCO_2$ maps generation, but in the entire section of methods, no paragraph or sentence was about EOF

[Responds]: The reviewer is correct that EOF was used to obtain the main spatiotemporal pattern of the RS derived $pCO_2$ and then as features in our reconstructed model. Following suggestions, we havel added the information of EOF as follows "The EOF reflects the spatial commonality of variables shown in the time series, thus it is widely used to calculate spatial patterns of climate variability (e.g. Levitus et al., 2005; Dye et al., 2020; McMonigal and Larson, 2022). Typically, the spatial commonality of variables, also named EOF modes, is found by computing the eigenvalues and eigenvectors of a spatially weighted anomaly covariance matrix of a field. Each EOF modes' corresponding variance represents its degree of interpretation of the spatial pattern of a variable." (Lines 245-249).

3. The language of the manuscript still need some efforts. The current version contains too many redundant phrases and sentences without clear meaning and very difficult to read through and get the logical flow. Readers expect concise and precise expression in an academic paper. and there are some grammar mistake and fuzzy expression.

[Responds]: We have paid special attention on the presentation during our revisions.

4. the range of legend in nearly all the map figures were too large and cannot show the spatial gradient of $pCO_2$ distribution, e.g, figure 6, 8, 11, 12,13.

[Responds]: Accepted. We have adjusted the range of colorbar in figures as follows (Figure R1-R7).

[Figure]

**Figure R1. Seasonal and monthly sea surface *p*CO₂ fields in the South China Sea. The data sources can be found in Table 1 (a. winter; b. December; c. January; d. February; e. Spring; f. March; g. April; h. May; i. Summer; j. June; k. July; l. August; m. Fall; n. September; o. October; p. November).**

[Figure]

**Figure R2. Reconstructed seasonal and annual sea surface $p$CO₂ fields in the South China Sea during the period 2003 to 2020 (a, 2003-2011; b, 2012-2020).**

[Figure]

**Figure R3. Differences between the seasonal and monthly reconstructed *p*CO₂ and the in situ *p*CO₂ data for the test set (a. winter; b. December; c. January; d. February; e. Spring; f. March; g. April; h. May; i. Summer; j. June; k. July; l. August; m. Fall; n. September; o. October; p. November). .**

[Figure]

**Figure R4. Difference between the reconstructed $pCO_2$ data and four independently in situ datasets during the four seasons. In (a), the numbers 1–4 represent September (2018.9, b), December 2018 (2018.12, c), August 2019 (2019.8, d), and April 2020 (2020.4, e), respectively.**

[Figure]

**Figure R5.** Uncertainties of the reconstructed $p$CO$_2$ fields (a, Total uncertainty in Equation 6; b. the first term of Equation 6; c. the second term of Equation 6; d stands for the $(\frac{\partial pCO2}{\partial SSS})dSSS$ in the the second term of Equation 6; e stands for the $(\frac{\partial pCO2}{\partial SST})dSST$ in the the second term of Equation 6; f stands for the $(\frac{\partial pCO2}{\partial Chl\ a})dChl\ a$ in the the second term of Equation 6; g stands for the $(\frac{\partial pCO2}{\partial RS\_derived\_pCO2})dRS\_derived\_pCO2$ in the the second term of Equation 6.

[Figure]

**Figure R6. Long-term (2003–2020) seasonal and monthly average $pCO_2$ field (unit: µatm) (a. winter; b. December; c. January; d. February; e. Spring; f. March; g. April; h. May; i. Summer; j. June; k. July; l. August; m. Fall; n. September; o. October; p. November).**

[Figure]

**Figure R7. Long-term (2003–2020) seasonal and monthly averaged $pCO_2$ field in the region north of 18°N (unit: μatm) (a. winter; b. December; c. January; d. February; e. Spring; f. March; g. April; h. May; i. Summer; j. June; k. July; l. August; m. Fall; n. September; o. October; p. November).**

5. what is the intention of including figure 4, if it is the quality of the remote sensing based $pCO_2$ maps included for further $pCO_2$ maps derivation, should the authors just need to include the information from the data distributor?

[Responds]: The reviewer is right that Figure 4 showed the quality of the RS-derived $pCO_2$ data. Following suggestions, we have removed this figure in the revision.

6. the study site section(2.1) should just serve the question "why mapping $pCO_2$ in SCS is important?", no other information is needed here.

[Responds]: We thank the reviewer for the comment. In the revision the importance of mapping $pCO_2$ in SCS have been added to section 2.1. The spatial distribution of $pCO_2$ is largely controlled by water mass missing and exchanges, thus, we have retained in the introduction to the surface ocean circulation and water mass exchanges in the South China Sea in this section.

7. be consistent with the terminology, sometimes it is "in-situ", but "observational data" and "observed data" were present many times.

[Responds]: Accepted. We have unified the 'in-situ'/'observational data'/'observed data' to 'in situ data'.

8. in the abstract (line 12-14,), the importance of mapping $pCO_2$ in SCS should be addressed before presenting the method, generated data and its quality.

[Responds]: Accepted. Before presenting our method, we have added the following information "The South China Sea (SCS) is the largest marginal sea in the North Pacific Ocean, where intensive field observations including mappings of sea-surface partial pressure of $CO_2$ ($pCO_2$) have been conducted over the last two decades. It is one of the most studied marginal seas in terms of carbon cycling and could thus be a model system for marginal sea carbon research." in lines 12-14 to show the importance of mapping $pCO_2$ in the SCS.

9. part of the input $pCO_2$ data of the presented study is from unpublished study (line 158), meaning not peer-reviewed.

[Responds]: We used two unpublished datasets in this paper. One of them is sea surface $pCO_2$ in China seas (0-42°N, 105-130°E) over 2003-2019 with a spatial resolution of 1 km and temporal resolution of a month (Bai et al., unpublished, line 158). This is the second version of $pCO_2$ in China seas. The first version was published on the SatCO2 website (http://www.satco2.com/index.php?m=content&c=index&a=show&catid=317&id=188) based on Bai et al. (2015). And this second version data can be cited as follows "Yu, S., Song, Z., Bai, Y., and He, X.: Remote Sensing based Sea Surface partial pressure of CO2 ($pCO_2$) in China Seas (2003-2019) (2.0). Zenodo, 2022. https://doi.org/10.5281/zenodo.7372479".

Another dataset is the SSS data produced by 'Wang et al (in press)' in line 212. This paper has been accepted by Remote Sensing and its DOI number will be added in the revision as "Wang, Z., Wang, G., Guo, X., Hu, J., and Dai, M. Reconstruction of High-Resolution Sea Surface Salinity over 2003–2020 in the South China Sea Using the Machine Learning Algorithm LightGBM Model. Remote. Sens., 14, 6147, 2022. https://doi.org/10.3390/rs14236147 ".

Thus, in the revision, we have updated the information accordingly.

10, line 308: Figure 7, validating the model output with the model training data gives no useful information, suggest removing this part

[Responds]: Accepted. In the revision, we have only kept the results of test sets in Figure 7 as follows (Figure R8).

[Figure]

**Figure R8. Comparison between the monthly reconstructed and the in situ pCO2 values for Tesing set (monthly results were overlaid to the four seasons: (a) Winter: Dec., Jan., Feb.; (b) Spring: Mar., Apr., May; (c) Summer: Jun., Jul., Aug.; (d) Fall: Sept., Oct., Nov.).**

Minor comments

line 15-17"Using a machine learning-based method facilitated by empirical orthogonal function (EOF)....   between 2003 and 2020" should specifically mention what kind of data was used for the methods input.

[Responds]: Accepted. Please refer to our response to Major Comment # 1 as of above.

linse 17- 20 "We validate our reconstruction with three independent testing datasets where,.... northern basin of the SCS." how independent are the three data set?

[Responds]: We validate our reconstruction with three independent testing datasets which are not involved model training. We have added this information in our revision (Line 20).

line 22 "our reconstructions and observed data" grammar mistake.
[Responds]: Accepted. In the revision, we have rewritten this sentence as follows " The root-mean-square error (RMSE) between our reconstructed data and in situ data in Test 1 averaged to ~10 μatm." (Lines 24-25).

Line 27-28 "we present a new method to assess the uncertainty that includes the bias from the reconstruction and its sensitivity to the features,... quantifies the spatial distribution patterns of uncertainty." then the assessment method should be concisely introduced here. in addition, given this is a data presentation paper, the newly developed method should not in the highlight, unless it is a method presentation paper.
[Responds]: In the revision, we have rewritten this sentence as follows "we assessed the uncertainty resulting from the bias of the reconstruction and its sensitivity to the features." (Lines 29-30).

line 19 "that our reconstruction is effectively captures the main features of both the" ,check the grammar.
[Responds]: We apologize for the mistake. In the revision, we have rewritten this sentence as follows "our reconstruction effectively captures the main spatial and temporal features of sea surface $pCO_2$ distributions in the SCS." (Lines 30-31).

line 38,, "22–26%",   I assume it should be 22%–26%.
[Responds]: The reviewer is correct, and we have made the correction in the revision (Line 41).

line 54-55:    ":The former typically use statistical interpolations and regression methods" does not fit with the neighouring sentence, rewrite it or delete it.
[Responds]: Accepted. In the revision, we have deleted this sentence.

line 61- 63 ,"However, because of the complex and dynamic nature of biogeochemical and physical processes in coastal areas, characterization of sea surface $pCO_2$ and subsequently the air-sea CO2 fluxes both in time and space in marginal seas remains challenging", this sentence is too strong and undermines the motivation of presented study, rewrite it,
[Responds]: Accepted. In the revision, we have rewritten this sentence as follows "Consequently, machine learning has increasingly become a routine approach in reconstruction of sea surface $pCO_2$ in open ocean regimes (e.g., Zeng et al., 2017; Li et al., 2019). However, it remains challenging to extend this method to marginal seas featuring more dynamic changes in both time and space." (Lines 61-63).

line 67: "clear need", what kind of need is clear need? a need can be strong, urgent, but not clear, need itself is a clear expression,
[Responds]: Accepted. In the revision, we have changed this sentence to "Therefore, there is a strong need to achieve surface water $pCO_2$ coverage in the SCS with spatiotemporal resolution as high as possible." (Lines 67-68).

line 73:   "(sea surface temperature, SST; chlorophyll a, Chl a),", pay attention to journal requirements on abbreviation
[Responds]: Accepted. In the revision, we have rewritten this sentence as follows "Zhu et al. (2009) presented an empirical approach to estimate sea surface $pCO_2$ in the northern SCS using remote sensing-derived (RS-derived) data including sea surface temperature (SST) and chlorophyll $a$ (Chl $a$), ..." (Lines 72-73).

line 74: "underway "pay attention to the usage of underway, it is ambiguous in the manuscript.

[Responds]: Accepted. In the revision, we have changed this to "in situ data" throughout the revision.

line 82, "the whole China Sea", where is the China Sea? do you mean all the seas in China's territory?
[Responds]: We referred to South China Sea, East China Sea, Yellow Sea, and Bohai Sea (99 - 130°E & 0 - 45°N). In the revision, we have added more details accordingly (Line 82).

line 84: "(reported in Wang et al., 2021).", pay attention to the format of the reference citation
[Responds]: Accepted. We have made this correction.

line 84: "Bai et al. (unpublished) subsequently", if the work is not publised, then it should not be cited or discussed, as it is not peer-reviewed.
[Responds]: This dataset is an updated version based on Bai et al. (2015). Please refer to our response to Major Comment # 9 as of above. In the revision, we have updated this citation to "Yu et al. (2022)".

line 94-96: include the input data here.
[Responds]: Accepted. In the revision, we have added some details of input data as follows "The input data in our reconstructed model include remote sensing-derived sea surface salinity, sea surface temperature, and chlorophyll, the spatial pattern of $pCO_2$ constrained by EOF, atmospheric $pCO_2$, and time labels (month)." (Lines 101-103).

line 137-138 : there is no asterisk in the table and the meaning of the asterisk led note is not clear.
[Responds]: Accepted. In the revision, we have modified this Table as follows (Table R1).

**Table R1. Summary of the seasonal in situ data of sea surface $pCO_2$ in the South China Sea for the period 2003-2020 used in this study.**

| Season | Spring | | | | Summer | |
| --- | --- | --- | --- | --- | --- | --- |
| | March | April | May | June | July | August |
| Cruise time | 2004.03 | 2005.04 2008.04 2009.04 2012.04 2020.04* | 2004.05 2011.05 2014.05 2020.05* | 2006.06 2016.06 2017.06* 2019.06* 2020.06* | 2004.07 2005.07 2007.07 2008.07 2009.07 2012.07 2015.07* 2019.07* | 2007.08 2008.08 2019.08* |

| Season | Fall | | | Winter | | |
| --- | --- | --- | --- | --- | --- | --- |
| | September | October | November | December | January | February |
| Cruise time | 2004.09 2007.09 2008.09 2020.09* | 2003.10 2006.10 | 2006.11 2010.11 | 2006.12 | 2009.01 2010.01 2018.01 | 2004.02 2006.02 |
| Data source | | | Li et al. (2020) *This study | | | |

line 144 "Figure 3 shows the spatial and temporal distributions of surface water $pCO_2$.", the spatial distribution of in-situ

measurements or data from other source?

[Responds]: Figure 3 shows the spatial distribution of in-situ measurements. In the revision, we have added more details as follows: "Figure 3 shows the spatial and temporal distributions of in situ sea surface $pCO_2$." (Line 153) .

line 157:  how the remote sensing-derived $pCO_2$ data were derived? which methods, what is the quality? and output from unpublished study should not be used.

[Responds]: This dataset is an updated version based on Bai et al. (2015). Please refer to our response to Major Comment # 9 as of above. In the revision, we have changed this citation to "Yu et al. (2022)", and also added more details of this dataset as follows "Yu et al. (2022) subsequently used a non-linear regression method to develop a retrieval algorithm for seawater $pCO_2$ in the China Seas, and the RS-derived $pCO_2$ data from 2003-2018 were provided by the SatCO$_2$ platform (www.SatCO2.com). In the retrieval algorithm of Yu et al. (2022), the input parameters include sea surface temperature, chlorophyll-a concentration, remote sensing reflectance of three bands (Rrs412, 443, 488 nm), the temperature anomaly in the longitude direction, and the theoretical thermodynamic background $pCO_2$ under corresponding SST. Although the RMSE associated with the RS-derived $pCO_2$ product was relatively large (21.1 μatm), it successfully showed major spatial patterns of the sea surface $pCO_2$ in the China Seas (Yu et al., 2022)." (Lines 84-90).

line 184-187: "Wang et al. (in preparation) found a relatively high differential between the....observed data", meaning of this super long sentence is not clear.

[Responds]: Accepted. In the revision, we have modified this sentence as follows "For sea surface salinity (SSS) data, Wang et al. (2022) found relatively large differences between different open source SSS databases (i.e., multi-satellite fusion data from https://podaac.jpl.nasa.gov/; model data from https://climatedataguide.ucar.edu/; multidimensional covariance model data from https://resources.marine.copernicus.eu/) and the in situ SSS data." (Lines 194-197).

line 198 "$pCO_2$ filling method of", should explain the filling method here!

[Responds]: Accepted. In the revision, we have modified this sentence as follows "Secondly, we filled missing $pCO_2$ measurements with the RS-derived $pCO_2$ data according to Fay et al. (2021) (see more details in Section 3.1)." in Lines 208-209 because that the $pCO_2$ filling method would be explained in section 3.1.

line 201: "$pCO_2$ reconstruction model"  $pCO_2$ reconstruction was used many times in the manuscript, but sea surface $pCO_2$ is not something one can reconstruct, it is a properties or variable of of the sea water, one can measure it ,describe it, retrieve its distribution, but not reconstruct $pCO_2$ itself.  So, please pay attention to the verb usage.

[Responds]: Accepted. In the revision, we have changed "$pCO_2$ reconstruction model" to "$pCO_2$ retrieval algorithm".

References

Dye, A. W., Rastogi, B., Clemesha, R. E. S., Kim, J. B., Samelson, R. M., Still, C. J., & Williams, A. P.: Spatial patterns and trends of summertime low cloudiness for the Pacific Northwest, 1996–2017. Geophysical Research Letters, 47, e2020GL088121, 2020. https://doi.org/10.1029/2020GL088121

Levitus, S., Antonov, J. I., Boyer, T. P., Garcia, H. E., and Locarnini, R. A.: EOF analysis of upper ocean heat content, 1956–2003, Geophys. Res. Lett., 32, L18607, 2005. doi:10.1029/2005GL023606.

McMonigal, K., & Larson, S. M.: ENSO explains the link between Indian Ocean dipole and Meridional Ocean heat transport. Geophysical Research Letters, 49, e2021GL095796, 2022. https://doi.org/10.1029/2021GL095796

Wang, Z., Wang, G., Guo, X., Hu, J., and Dai, M. Reconstruction of High-Resolution Sea Surface Salinity over 2003–2020 in the South China Sea Using the Machine Learning Algorithm LightGBM Model. Remote. Sens., 14,

6147, 2022. https://doi.org/10.3390/rs14236147.

Yu, S., Song, Z., Bai, Y., and He, X.: Remote Sensing based Sea Surface partial pressure of CO2 (pCO2) in China Seas (2003-2019) (2.0). Zenodo, 2022. https://doi.org/10.5281/zenodo.7372479.

---

## Author Response (AR2)

**[Response to Editor]**

Dear authors,

as you could see from the evaluations, your work has significant shortcomings in completeness, quality of presentation and quality of data. This last one is a very important requirement for ESSD. I therefore ask you to re-submit a new version of the work, improving it, eliminating weak points. The article will then be evaluated again.

Regards

[Response]:

We really appreciate the constructive comments from the two reviewers and the editor.

In the terms of completeness of paper, the reviewers pointed out that the data introduction section needed more details, and the language of the manuscript still needed some effort. Thus, we have supplemented the information for the data accordingly, and paid special attention to its presentation in our revised manuscript.

In the terms of the presentation, Reviewer#1 commented that Figures 5, 8 and 11 need to be modified, and Reviewer#2 pointed out the range of the legend in nearly all the map figures was too large. According to these comments, we have re-made Figures 5, 8 and 11 in the revision to meet the requirements of Reviewer#1. Moreover, to more clearly show the spatial gradient of the $pCO_2$ distribution, we have adjusted the range of the color bar in all applicable figures in the revision.

In the terms of the data quality, Reviewer#1 was mainly concerned whether the algorithms will provide the same accuracy during different seasons when different features become more important. We have addressed this comment, and performed complementary experiments for the other three seasons in the revision. Reviewer#2 pointed out that some of the input $pCO_2$ data in the current study is from unpublished works, for example data from a sea surface $pCO_2$ dataset in the China seas (0-42°N, 105-132°E) from 2003-2019 with a spatial resolution of 1 km and temporal resolution of a month (Bai et al., unpublished, line 158), and also a salinity dataset produced by 'Wang et al (in press)' in line 212. According to this comment, we have updated the information. The first version of RS-derived sea surface $pCO_2$ dataset was published on the SatCO2 website (http://www.satco2.com/index.php?m=content&c=index&a=show&catid=317&id=188) based on Bai et al. (2015), and the second version which we used in this paper can be cited as follows "Yu, S., Song, Z., Bai, Y., and He, X.: Remote Sensing based Sea Surface partial pressure of $CO_2$ ($pCO_2$) in China Seas (2003-2019) (2.0). Zenodo, 2022. https://doi.org/10.5281/zenodo.7372479". The SSS data produced by 'Wang et al (in press)' has been accepted by Remote Sensing and its DOI number has been added in the revision as "Wang, Z., Wang, G., Guo, X., Hu, J., and Dai, M. Reconstruction of High-Resolution Sea Surface Salinity over 2003–2020 in the South China Sea Using the Machine Learning Algorithm LightGBM Model. Remote. Sens., 14, 6147, 2022. https://doi.org/10.3390/rs14236147 ".

However, besides these major concerns, both reviewers gave relatively positive comments to our research and data. Reviewer#1 stated "The results show a good agreement with validation data and independent observations. ", and Reviewer#2 said "the study and the data it present is very meaningful.". Therefore, after our carefully edited revision, we are confident that the quality of our data fully meets the publication requirements of the journal.

**[Response to Reviewer#1]**

Spatial reconstruction of long-term (2003-2020) sea surface $p\text{CO}_2$ in the South China Sea using a machine learning based regression method aided by empirical orthogonal function analysis.

Authors presented a machine learning approach to reconstruct ocean $p\text{CO}_2$ over the South China Sea using the new drivers based on EOFs of Remote Sensing-derived $p\text{CO}_2$. These new drivers contribute to the estimation accurate $p\text{CO}_2$ product at high spatial resolution. The final product represents a monthly 0.05°x0.05°surface ocean $p\text{CO}_2$ for the period 2003-2020. The results show a good agreement with validation data and independent observations. Authors discussed the seasonal effect on the reconstruction and mentioned seasonal processes that can affect the ocean $p\text{CO}_2$. One of the interesting points in this work is the estimation of uncertainties. Authors introduced the estimation of uncertainties from features used in $p\text{CO}_2$ reconstruction. The article is well structured, and it is easy to follow.

However, I found that the article missed the clarity and not all important details are presented or well explained. Below, I listed points that need to be improved and clarified before publication.

[Response]: We thank the reviewer for the positive comments. We have listed our point-by-point responses below.

Comments:

- The description and correct definition of data used. In your study you use the data from field survey that you call "observations" or "observed data". Also, you use remote sensing-derived data. However, it is not clear that the data from remote sensing is not direct measurements of $p\text{CO}_2$, and it is derived product as you mentioned in 2.3 (line 156). In you abstract you speak about the comparison between "the remote sensing and observed data" (line 23) that is ambiguous. The remote sensing data are observations too and it is not exactly what was used in the paper as it was derived product. I suggest you call the data from filed survey "in situ data", and call the data derived from remote sensing "remote sensing-derived data" everywhere in the manuscript.

[Response]: The reviewer is right that remote sensing is also an observation tool. Revisions have been made throughout the manuscript.

- Please add more details about how and what exactly was measured during the field survey. Is it the surface fugacity of CO2? If yes, you need to mention it and precise that you estimate $p\text{CO}_2$ from fugacity.

[Response]: We thank the reviewer for the suggestion. The details of the in situ $p\text{CO}_2$ data collected were previously described in Li et al. (2020). During most cruises, $p\text{CO}_2$ was measured continuously with a non-dispersive infrared spectrometer (Li-Cor® 7000) or by Cavity Ring-Down Spectroscopy (Picarro G2301) integrated in a GO-8050 system (General Oceanic Inc. USA) onboard the research vessels. We have added the following information to our revision "During the cruises, sea surface $p\text{CO}_2$ was measured underway. The measurement and data processing followed the SOCAT (Surface Ocean CO2 Atlas) protocol (Li et al., 2020)." (Lines 133-135).

- Please add more details on how remote sensing-derived data were produced. The website you cite in your paper www.SatCO2.com shows only homepage and it is impossible to navigate as all other webpages where we could find details about the product is forbidden. There is a little description of the product in introduction (lines 80-86), however, there is no indication that this product will be used further in the article.

[Response]: We have added the following information to show how remote sensing (RS)-derived data were produced: "Yu et al. (2022) subsequently used a non-linear regression method to develop a retrieval algorithm for seawater $p\text{CO}_2$ in the China Seas, and the RS-derived $p\text{CO}_2$ data from 2003-2018 were provided by the SatCO2 platform

(www.SatCO2.com). In the retrieval algorithm of Yu et al. (2022), the input parameters include sea surface temperature, chlorophyll-a concentration, remote sensing reflectance of three bands (Rrs412, 443, 488 nm), the temperature anomaly in the longitudinal direction, and the theoretical thermodynamic background $p\text{CO}_2$ under corresponding SST. Although the RMSE associated with the RS-derived $p\text{CO}_2$ product was relatively large (21.1 μatm), it successfully showed the major spatial patterns of sea surface $p\text{CO}_2$ in the China Seas (Yu et al., 2022)." (Lines 81-86).

In the revision we have also added the following information "Wang et al. (2021) demonstrate that the spatial modes of RS-derived data calculated using the EOF are effective in providing spatial constraints on the data reconstruction and are thus adopted in this study." in lines 93-95 to explain how the RS-derived $p\text{CO}_2$ data were used in this study.

\- Please make corresponding changes in Figure 5: observed data to in situ data; RS $p\text{CO}_2$ data to RS-derived $p\text{CO}_2$ data. As you use SSS data reconstructed using ML it is incorrect to put it together with observed SST and Chl-a, or you should precise it in your figure like "ML SSS".

[Response]: We acknowledge this comment, and have modified Figure 5 accordingly (Figure R1). We note that the SSS data from 2003-2020 in the South China Sea used in the present study were reconstructed based on the MODIS-Aqua remote sensing data (Wang et al., 2022). We have added this information in our revision in Lines 186-189.

[Figure]

**Figure R1. Procedure for the reconstruction of sea surface $p$CO₂ using machine learning. RS-derived data = remote sensing-derived data, RMSE = root mean square error, MAPE= mean absolute percentage error, and R² = coefficient of determination, and MAE = mean absolute error.**

- Please add more information on the datasets that you introduced in lines 150-152.

[Response]: We have added more details as follows: "In addition to the above in situ sea surface $p$CO₂ data, to verify the accuracy of our reconstruction model in extrapolation to periods lacking training datasets, we selected the in situ sea surface $p$CO₂ data collected in four independent surveys corresponding to the four seasons: September 2018 (fall), December 2018 (winter), August 2019 (summer), and April 2020 (spring). Furthermore, we used another dataset of sea surface $p$CO₂ calculated from observed dissolved inorganic carbon and total alkalinity during 2003–2019 at the Southeast Asia Time-Series (SEATs) station (data from Dai et al., 2022) to test the long-term consistency of the reconstruction." in Lines 153-158.

- Figures' captions. Please add more information in figures' captions. Each subplot needs to be introduced in the caption.

[Response]: Agreed. We have introduced each subplot in the revised manuscript accordingly.

- Tables. Please keep same number of digits in fractional part for your results in tables: Table 2, Summer RMSE has 3 digits while all other values limited by 2 digits in fractional part. Also, please use the same numbers in the text and in tables, line 163.

[Response]: Accepted. We have retained 2 significant digits after the decimal point in the revised manuscript.

- Abbreviations. Please define abbreviations when you use it for the first time: for example, SSS in line 184.

[Response]: Accepted. SSS stands for the sea surface salinity. In the revision, we have defined all abbreviations when they first appear

- Verification of different regression algorithms. Lines 255-261. To test the capacity of different algorithm you choose the summer season due to its "greatest temporal sampling coverage". However, we can see in your article that there is a strong seasonality in $pCO_2$ distribution. How can you be sure that algorithms will provide the same accuracy during different seasons when other features can become more important?

[Response]: The reviewer is correct that we performed complementary experiments for the other three seasons, showing that the differences resulting from different algorithms during the other seasons was minor (<2 µatm in RMSE, Table R1).

**Table R1. RMSE associated with different algorithms in different seasons.**

| Season | Random Forest | LightGBM | CATBOOST | Multi-linear regression (Wang et al., 2021) |
|--------|---------------|----------|----------|---------------------------------------------|
| Spring | 10.65 µatm | 9.52 µatm | 8.17 µatm | NaN* |
| Summer | 26.53 µatm | 27.83 µatm | 16.15 µatm | 20.13 µatm |
| Fall | 10.34 µatm | 11.56 µatm | 10.35 µatm | NaN |
| Winter | 12.48 µatm | 12.75 µatm | 11.52 µatm | NaN |

**\*NaN stands for missing values**

In the revision, we have added Table R1 into the MS along with the following information: "From the above options, we chose three ensemble learning algorithms as the machine learning-based regression portion, and multi-linear regression methods (Wang et al., 2021) as the linear regression portion. We then used the K-fold and cross validation methods to verify the applicability of different regression algorithms in the $pCO_2$ reconstruction for seasonal training data. The results show that in summer the CATBOOST algorithm yields the best degree of accuracy with an RMSE of 16 µatm (Table R1). In contrast, the RMSE of LightGBM was 27 µatm, and that of Random Forest was 26 µatm. The RMSE was nearly 20 µatm using the linear regression algorithm employed by Wang et al. (2021). Thus, CATBOOST appears to provide a reliable algorithm for reconstructing $pCO_2$. In the other three seasons, however, different algorithms resulted in minor differences (~2 µatm in RMSE)." in Lines 265-272.

- Uncertainties. The method to estimate uncertainties should be presented in section 3.4 and not in the section where you discuss your results. In part 1 of equation 6 the function MAX does not do anything as you apply it to a scalar. What is $pCO_2$_recon in this equation? Does the part 2 of equation 6 represent the sum over the features? Do you base your estimation on the error propagation method (absolute/relative error of a function)? It would be interesting to see the effect of individual features on $pCO_2$ uncertainties and identify the feature that brings larger bias.

[Response]: Following suggestions, we have moved the method estimating uncertainties to section 3.5 and modified Equation 6 as follows (Equation R1) in the revision. Figure 11 has also been modified to Figure R2 to identify the uncertainty caused by each feature in the revision.

$$Uncertainty = MAX([\frac{\sum_{i=1,j=1,k=1}^{n}\frac{|OR\_Monthly\_Data(i,j,k)-Obs\_Monthly\_Data(i,j,k)|}{Obs\_Monthly\_Data(i,j,k)}}{num(i)+num(j)}, \dots , \frac{\sum_{i=1,j=1,k=n}^{n}\frac{|OR\_Monthly\_Data(i,j,k)-Obs\_Monthly\_Data(i,j,k)|}{Obs\_Monthly\_Data(i,j,k)}}{num(i)+num(j)}]) *$$

$$100\% * pCO2\_recon$$

$$+ \qquad \sum_{i=1}^{n}(\frac{\partial pCO2}{\partial Feature_i})dFeature_i \qquad . \qquad (R1)$$

[Figure]

**Figure R2. Uncertainties of the reconstructed sea surface $pCO_2$ fields (a, Total uncertainty in Equation 6; b. the first term of Equation 6; c. the second term of Equation 6; d represents $(\frac{\partial pCO2}{\partial SSS})dSSS$ in the the second term of Equation 6; e represents $(\frac{\partial pCO2}{\partial SST})dSST$ in the the second term of Equation 6; f represents $(\frac{\partial pCO2}{\partial Chl\,a})dChl\,a$ in the the second term of Equation 6; g stands for $(\frac{\partial pCO2}{\partial RS\_derived\_pCO2})dRS\_derived\_pCO2$ in the the second term of Equation 6.**

For the first term in Equation R1, $k$ stands for the $k$th month, $OR\_Monthly\_Data(i,j,k)$ stands for the $k$th monthly reconstructed data at longitude($i$) and latitude($j$), and $Obs\_Monthly\_Data(i,j,k)$ stands for the $k$th monthly in situ data

at longitude (*i*) and latitude (*j*). Therefore, the *MAX* in first term stands for that the maximum value between the *k* monthly bias ratios. And '$pCO_2\_recon$' stands for the reconstructed $CO_2$ data.

The second term in Equation R1 represents the sum of the features. According to Equation R1, the bias of RS-derived $pCO_2$ used in the second term of Equation R1 is ~21 μatm (Table 2), the bias of SST is ~ 0.27° (Qin et al., 2014), the bias of SSS is ~0.33 (Wang et al., 2022), and the bias of Chl-a is ~115% (Zhang et al., 2006). The results can be found in Fig. R1. The overall uncertainty (Fig. R1 a) is greater in the coastal area (~13 μatm) than in the basin (~10 μatm), and this spatial pattern is mainly determined by the second term. The spatial distribution of the first term in Equation R1 (Fig. R1 b) calculated from a "max bias ratio" is consistent with that of $pCO_2$. The second term in Equation R1 (Fig. R1 c) is calculated from the propagation of bias of each variable. The bias from Chl *a* (Fig. R1 f) shows the greatest effect on the reconstruction of all features. Although the bias of RS-derived $pCO_2$ is relatively large, the final influence of its bias on the reconstructed model results is negligible due to the EOF method (Fig. R1 g).

We have included this description of uncertainty in Section 4.3 of the revision.

- Conclusion. Line 424, please specify which machine learning method. Line 426, please specify that you used remote sensing-derived data.

[Response]: Accepted. The machine learning method is CATBOOST, and the input data we used in machine learning includes remote sensing-derived data (sea surface salinity, sea surface temperature, chlorophyll), the spatial patterns of $pCO_2$ calculated by the Empirical Orthogonal Function (EOF), atmospheric $CO_2$, and time-labels (month). We have specified this information in the revision.

- Data pre-processing. Are the data used in ML method pre-processed: interpolated on the same grid, normalized?

[Response]: Yes, all the data used in ML were interpolated on the same grid. In the revision, we have added this information: "Note that all the data used in machine learning have been interpolated on the same grid." in Line 214.

- Line 148: "relatively low $pCO_2$", what does it mean, how low is it?

[Response]: "relatively low $pCO_2$" means ~350 μatm. We have added this info in the revision (Line 151).

- Line 164: "current algorithm", please precise, what algorithm are you talking about.

[Response]: It refers to the mechanic semi-analytical algorithm (MeSAA) and non-linear regression. In the revision, we have added this information (Line 170) .

- Line 187: "our observed data", please precise which data.

[Response]: "our observed data" stands for the in situ data. In the revision, we have made modifications accordingly.

- You should mention in section 2.3 that there is a section 3.1 where you explain how you fill missing points in RS-derived $pCO_2$ product.

[Response]: Accepted. We have mentioned this information at the end of section 2.3.

- Could you please provide a figure to show the distribution of training samples you mentioned in lines 201-202: "To ensure that the model had sufficient training samples in the coastal area, we divided the entire SCS into two regions along the 200 m depth contour."

[Response]: Accepted, and such a figure (Figure R3) has been added in the revision as Figure 5.

[Figure]

**Figure R3. Spatial distribution of training samples (a) and testing samples (b). The black dashed line stands for the 200m depth contour.**

- Figure 8: It is difficult to see the results for test set. The results for training set look very similar and homogeneous, I would suggest keep only test set here.

[Response]: Accepted, and we have only kept the results of test sets in Figure 8 as shown in Figure R4.

[Figure]

**Figure R4. Differences between reconstructed seasonal and monthly $p\mathrm{CO_2}$ and the in situ $p\mathrm{CO_2}$ for the test set (a. winter; b. December; c. January; d. February; e. Spring; f. March; g. April; h. May; i. Summer; j. June; k. July; l. August; m. Fall; n. September; o. October; p. November).**

- Line 322: "The greatest bias occurs in the Pearl River plume area in summer". Could you please indicate how large is this bias?

[Response]: It is about 35 µatm. We have added this information in the revision (Line 352).

- Line 323: what is tpCO2?

[Response]: It should be 'the $p\mathrm{CO_2}$', and we have removed this typo in the revision.

- Line 376: you say here that "the lowest value occurs in January", in the next sentence you say "$p\mathrm{CO_2}$ first decreases in December and then increases in January". It means that the lowest value is in December. Please clarify it.

[Response]: It is a typo, "$p\mathrm{CO_2}$ decreases in December and then increases in January" should be "then increases after January". The lowest value is in January. In the revision, we have made these corrections.

- Line 417: "…a source or sink of atmospheric CO2 is influenced by seasonal changes and physical processes". Please specify seasonal changes and physical processes.

[Response]: Accepted. We have added more details as follows in the revision (Lines 455-458).: "Subregion B can be a significant sink of atmospheric $CO_2$ as demonstrated by its low sea surface $pCO_2$ when the Pearl River plume spreads more widely in summer. In contrast, in winter when the Kuroshio intrusion is strong, both Subregions B and D have high sea surface $pCO_2$, indicating both subregions are sources of atmospheric $CO_2$."

Typo and style:

Line 15: I would suggest using word "sparse" instead of "incomplete". Line 37: Please change "…annually mitigates…" to "…annually mitigated…" as you refer to the concrete period of 1960-2019; or change the sentence completely.

[Response]: Accepted and revised as suggested(Line 15).

Line 50: "Numerical ocean models of performance.." Please remove "of performance".

[Response]: Accepted and revised as suggested (Line 50).

Line 53: I would not use the word "alternative". The data-based approaches are different methods to study ocean biogeochemistry that can be complementary to biogeochemical models.

[Response]: Accepted. We have rewritten this sentence as follows "data-based approaches have become an important complement to numerical models" (Lines 54).

Line 119: "CCC, yellow line in Fig. 1". There is no yellow line in Fig. 1. CCC corresponds to the green line.

[Response]: The reviewer is correct, and we have made the corrections.

Fig. 2: Please change the name of your colorbar to "number of data".

[Response]: Accepted. We have changed the name of our colorbar in Fig.2 to "number of data"

Line 145: "Spatially, the $pCO_2$ distribution in the basin is relatively homogeneous, but shows large variability in the northern region". I suppose you meant "Spatially, the $pCO_2$ distribution in the basin is relatively homogeneous with large variability in the northern region".

[Response]: Accepted. We have rewritten this sentence as your suggestion (Lines 148-149).

Line 288: Please change "the continuity changes.." to "the continuous changes".

[Response]: Accepted. We have changed "the continuity changes" to "the continuous changes" (Line 317).

Line 300: Please add that these estimations are over the seasons.

[Response]: Accepted. We have added this information in the revision.

Line 322: Please change "The greatest bias" to "The largest bias".

[Response]: Accepted. We have changed "The greatest bias" to "The largest bias" (Line 352).

Line 358: Please change "Equation (7)" to "Equation (6)".

[Response]: Accepted. We have made these corrections in the revision.

Line 408: Missing space between "uncertainty" and "is".

[Response]: Accepted. We have removed this typo in the revision (Line 390).

References

Wang, Z., Wang, G., Guo, X., Hu, J., and Dai, M. Reconstruction of High-Resolution Sea Surface Salinity over 2003–2020 in the South China Sea Using the Machine Learning Algorithm LightGBM Model. Remote. Sens., 14, 6147, 2022. https://doi.org/10.3390/rs14236147.

Yu, S., Song, Z., Bai, Y., and He, X.: Remote Sensing based Sea Surface partial pressure of $CO_2$ ($pCO_2$) in China Seas (2003-2019) (2.0). Zenodo, 2022. https://doi.org/10.5281/zenodo.7372479.

**[Response to Reviewer#2]**

The presented study aimed to produce monthly sea surface $p\mathrm{CO_2}$ maps for the South China Sea (SCS). Given SCS is a typical temperate/subtropical marginal sea, the $p\mathrm{CO_2}$ sea surface maps for this waters is necessary for understanding the CO2 flux in temperate marginal sea and even global CO2 flux. From this perspective, the study and the data it present is very meaningful. However, the manuscript still have some major flaws which do not advise me to give a yes to publishing it in its current status.

[Response]: We appreciate that the reviewer valued our study. Our point-by-point responses are listed below.

Major comments

1. The manuscript was about a dataset generation, but from the abstract and the last section, what kind of data was used as input for the method was missing.

[Response]: Accepted. We have added the information for input data in the abstract and in the last section of the revision. Note that the data input includes remote sensing-derived data (sea surface salinity, sea surface temperature, chlorophyll), the spatial pattern of $p\mathrm{CO_2}$ calculated by the Empirical Orthogonal Function, atmospheric $\mathrm{CO_2}$, and time-labels (month).

2. As I understand EOF was an important part of the method used for $p\mathrm{CO_2}$ maps generation, but in the entire section of methods, no paragraph or sentence was about EOF

[Response]: The reviewer is correct that an EOF was used to obtain the main spatiotemporal pattern of the RS-derived $p\mathrm{CO_2}$ and then as features in our reconstructed model. Following the reviewer's suggestions, we have added the information of the EOF as follows "The EOF reflects the spatial commonality of variables shown in the time-series, thus it is widely used to calculate spatial patterns of climate variability (e.g. Levitus et al., 2005; Dye et al., 2020; McMonigal and Larson, 2022). Typically, the spatial commonality of variables (EOF modes), is found by computing the eigenvalues and eigenvectors of a spatially weighted anomaly covariance matrix of a field. Each EOF modes' corresponding variance represents its degree of interpretation of the spatial pattern of a variable." (Lines 237-241).

3. The language of the manuscript still need some efforts. The current version contains too many redundant phrases and sentences without clear meaning and very difficult to read through and get the logical flow. Readers expect concise and precise expression in an academic paper. and there are some grammar mistake and fuzzy expression.

[Response]: We have paid special attention on the presentation during our revisions.

4. the range of legend in nearly all the map figures were too large and cannot show the spatial gradient of $p\mathrm{CO_2}$ distribution, e.g, figure 6, 8, 11, 12,13.

[Response]: Accepted. We have adjusted the range of the colorbar in figures as follows (Figure R1-R7).

[Figure]

**Figure R1. Seasonal and monthly sea surface $p$CO$_2$ fields in the South China Sea. The data sources can be found in Table 1 (a. Winter; b. December; c. January; d. February; e. Spring; f. March; g. April; h. May; i. Summer; j. June; k. July; l. August; m. Fall; n. September; o. October; p. November).**

[Figure]

**Figure R2. Reconstructed seasonal and annual sea surface $p$CO$_2$ fields in the South China Sea during the period of 2003 to 2020 (a, 2003-2011; b, 2012-2020).**

[Figure]

**Figure R3. Differences between the seasonal and monthly reconstructed *p*CO₂ and the in situ *p*CO₂ data for the test set (a. Winter; b. December; c. January; d. February; e. Spring; f. March; g. April; h. May; i. Summer; j. June; k. July; l. August; m. Fall; n. September; o. October; p. November). .**

[Figure]

**Figure R4. Difference between the reconstructed $p$CO₂ data and four independent in situ datasets during the four seasons. In (a), the numbers 1–4 represent September (2018.9, b), December 2018 (2018.12, c), August 2019 (2019.8, d), and April 2020 (2020.4, e), respectively.**

[Figure]

**Figure R5.** Uncertainties of the reconstructed $pCO_2$ fields (a, Total uncertainty in Equation 6; b. the first term of Equation 6; c. the second term of Equation 6; d represents $(\frac{\partial pCO2}{\partial SSS})dSSS$ in the the second term of Equation 6; e represents $(\frac{\partial pCO2}{\partial SST})dSST$ in the the second term of Equation 6; f represents $(\frac{\partial pCO2}{\partial Chl\,a})dChl\,a$ in the the second term of Equation 6; g represents $(\frac{\partial pCO2}{\partial RS\_derived\_pCO2})dRS\_derived\_pCO2$ in the the second term of Equation 6.

[Figure]

**Figure R6. Long-term (2003–2020) seasonal and monthly average *p*CO₂ fields (unit: µatm) (a. Winter; b. December; c. January; d. February; e. Spring; f. March; g. April; h. May; i. Summer; j. June; k. July; l. August; m. Fall; n. September; o. October; p. November).**

[Figure]

**Figure R7. Long-term (2003–2020) seasonal and monthly averaged $pCO_2$ fields in the region north of 18°N (unit: µatm) (a. Winter; b. December; c. January; d. February; e. Spring; f. March; g. April; h. May; i. Summer; j. June; k. July; l. August; m. Fall; n. September; o. October; p. November).**

5. what is the intention of including figure 4, if it is the quality of the remote sensing based $pCO_2$ maps included for further $pCO_2$ maps derivation, should the authors just need to include the information from the data distributor?

[Response]: The reviewer is right that Figure 4 showed the quality of the RS-derived $pCO_2$ data. Following suggestions, we have removed this figure in the revision.

6. the study site section(2.1) should just serve the question "why mapping $pCO_2$ in SCS is important?", no other information is needed here.

[Response]: We thank the reviewer for this comment. In the revision the importance of mapping $pCO_2$ in SCS has been added to section 2.1. The spatial distribution of $pCO_2$ is largely controlled by water mass mixing and exchanges, and thus we have retained in the introduction descriptions of the surface ocean circulation and water mass exchanges in the South China Sea in this section.

7. be consistent with the terminology, sometimes it is "in-situ", but "observational data" and "observed data" were present many times.

[Response]: Accepted. We have unified the 'in-situ'/'observational data'/'observed data' to 'in situ data'.

8. in the abstract (line 12-14,), the importance of mapping $pCO_2$ in SCS should be addressed before presenting the method, generated data and its quality.

[Response]: Accepted. Before presenting our method, we have added the following information in lines 12-14 to show the importance of mapping $pCO_2$ in the SCS: "The South China Sea (SCS) is the largest marginal sea in the North Pacific Ocean, where intensive field observations including mappings of sea-surface partial pressure of $CO_2$ ($pCO_2$) have been conducted over the last two decades. It is one of the most studied marginal seas in terms of carbon cycling and could thus be a model system for marginal sea carbon research."

9. part of the input $pCO_2$ data of the presented study is from unpublished study (line 158), meaning not peer-reviewed.

[Response]: We used two unpublished datasets in this paper. One of them is RS derived sea surface $pCO_2$ in the China seas (0-42°N, 105-130°E) from 2003-2019 with a spatial resolution of 1 km and temporal resolution of one month (Bai et al., unpublished, line 158). The first version of them is sea surface $pCO_2$ dataset published on the SatCO2 website (http://www.satco2.com/index.php?m=content&c=index&a=show&catid=317&id=188) based on Bai et al. (2015), and the second version which we used in this paper can be cited as follows "Yu, S., Song, Z., Bai, Y., and He, X.: Remote Sensing based Sea Surface partial pressure of $CO_2$ ($pCO_2$) in China Seas (2003-2019) (2.0). Zenodo, 2022. https://doi.org/10.5281/zenodo.7372479".

Another dataset is the SSS data produced by 'Wang et al (in press)' in line 212. This paper has been on-line published in Remote Sensing and its DOI number is added in the revision as "Wang, Z., Wang, G., Guo, X., Hu, J., and Dai, M. Reconstruction of High-Resolution Sea Surface Salinity over 2003–2020 in the South China Sea Using the Machine Learning Algorithm LightGBM Model. Remote. Sens., 14, 6147, 2022. https://doi.org/10.3390/rs14236147 ".

Thus, in the revision, we have updated the information accordingly.

10, line 308: Figure 7, validating the model output with the model training data gives no useful information, suggest removing this part

[Response]: Accepted. In the revision, we have only kept the results of test sets in Figure 7 as follows (Figure R8).

[Figure]

**Figure R8. Comparison between the monthly reconstructed and the in situ $pCO_2$ values for Testing set (monthly results were grouped into the four seasons: (a) Winter: December, January, Feb.; (b) Spring: March, April, May; (c) Summer: June, Jule, August; (d) Fall: September, October, November).**

Minor comments

line 15-17"Using a machine learning-based method facilitated by empirical orthogonal function (EOF).... between 2003 and 2020" should specifically mention what kind of data was used for the methods input.

[Response]: Accepted. Please refer to our response to Major Comment # 1 above.

linse 17- 20 "We validate our reconstruction with three independent testing datasets where,.... northern basin of the SCS." how independent are the three data set?

[Response]: We validate our reconstruction with three independent testing datasets which are not involved in model training. We have added this information to our revision (Line 20).

line 22 "our reconstructions and observed data" grammar mistake.

[Response]: Accepted. In the revision, we have rewritten this sentence as follows " The root-mean-square error (RMSE) between our reconstructed data and in situ data in Test 1 averaged ~10 μatm." (Lines 24-25).

Line 27-28 "we present a new method to assess the uncertainty that includes the bias from the reconstruction and its sensitivity to the features,... quantifies the spatial distribution patterns of uncertainty." then the assessment method should be concisely introduced here. in addition, given this is a data presentation paper, the newly developed method should not in the highlight, unless it is a method presentation paper.

[Response]: In the revision, we have rewritten this sentence as follows "we assessed the uncertainty resulting from the bias of the reconstruction and its sensitivity to the features." (Lines 29).

line 19 "that our reconstruction is effectively captures the main features of both the" ,check the grammar.

[Response]: We apologize for the mistake. In the revision, we have rewritten this sentence as follows "our reconstruction effectively captures the main spatial and temporal features of sea surface $pCO_2$ distributions in the SCS." (Lines 30-31).

line 38,, "22–26%", I assume it should be 22%–26%.

[Response]: Accepted and revised as suggested (Line 41).

line 54-55: ":The former typically use statistical interpolations and regression methods" does not fit with the neighouring sentence, rewrite it or delete it.

[Response]: Accepted. In the revision, we have deleted this sentence.

line 61- 63 ,"However, because of the complex and dynamic nature of biogeochemical and physical processes in coastal areas, characterization of sea surface $pCO_2$ and subsequently the air-sea CO2 fluxes both in time and space in marginal seas remains challenging", this sentence is too strong and undermines the motivation of presented study, rewrite it,

[Response]: Accepted. In the revision, we have rewritten this sentence as follows "Consequently, machine learning has increasingly become a routine approach in reconstructing sea surface $pCO_2$ in open ocean regimes (e.g., Zeng et al., 2017; Li et al., 2019). However, it remains challenging to extend this method to marginal seas featuring more dynamic changes in both time and space." (Lines 61-64).

line 67: "clear need", what kind of need is clear need? a need can be strong, urgent, but not clear, need itself is a clear expression,

[Response]: Accepted. In the revision, we have changed this sentence to "Therefore, there is a strong need to achieve surface water $pCO_2$ coverage in the SCS with spatiotemporal resolution as high as possible." (Lines 67-70).

line 73: "(sea surface temperature, SST; chlorophyll a, Chl a),", pay attention to journal requirements on abbreviation

[Response]: Accepted. In the revision, we have rewritten this sentence as follows "Zhu et al. (2009) presented an empirical approach to estimate sea surface $pCO_2$ in the northern SCS using remote sensing-derived (RS-derived) data including sea surface temperature (SST) and chlorophyll $a$ (Chl $a$), ..." (Lines 71-72).

line 74: "underway "pay attention to the usage of underway, it is ambiguous in the manuscript.

[Response]: Accepted. In the revision, we have changed this to "in situ data" throughout the revision.

line 82, "the whole China Sea", where is the China Sea? do you mean all the seas in China's territory?

[Response]: We referred to the South China Sea, East China Sea, Yellow Sea, and Bohai Sea (99 - 130°E & 0 - 45°N). In the revision, we have added more details accordingly (Line 82).

line 84: "(reported in Wang et al., 2021).", pay attention to the format of the reference citation

[Response]: Accepted. We have made this correction.

line 84: "Bai et al. (unpublished) subsequently", if the work is not publised, then it should not be cited or discussed, as it is not peer-reviewed.

[Response]: This dataset is an updated version based on Bai et al. (2015). Please refer to our response to Major Comment # 9 above. In the revision, we have updated this citation to "Yu et al. (2022)".

line 94-96: include the input data here.

[Response]: Accepted. In the revision, we have added some details of input data as follows "The input data in our reconstructed model include remote sensing-derived sea surface salinity, sea surface temperature, and chlorophyll, the spatial pattern of $pCO_2$ constrained by the EOF, atmospheric $pCO_2$, and time-labels (month)." (Lines 97-99).

line 137-138 : there is no asterisk in the table and the meaning of the asterisk led note is not clear.

[Response]: Accepted. In the revision, we have modified this Table as follows (Table R1).

**Table R1. Summary of the seasonal in situ sea surface $pCO_2$ data in the South China Sea from 2003-2020, used in this study.**

| Season | Spring | | | Summer | | |
|---|---|---|---|---|---|---|
| | March | April | May | June | July | August |
| Cruise time | 2004.03 | 2005.04 2008.04 2009.04 2012.04 2020.04* | 2004.05 2011.05 2014.05 2020.05* | 2006.06 2016.06 2017.06* 2019.06* 2020.06* | 2004.07 2005.07 2007.07 2008.07 2009.07 2012.07 2015.07* 2019.07* | 2007.08 2008.08 2019.08* |

| Season | Fall | | | | Winter | |
|---|---|---|---|---|---|---|
| | September | October | November | December | January | February |
| Cruise time | 2004.09 2007.09 2008.09 2020.09* | 2003.10 2006.10 | 2006.11 2010.11 | 2006.12 | 2009.01 2010.01 2018.01 | 2004.02 2006.02 |
| Data source | Li et al. (2020) *This study | | | | | |

line 144 "Figure 3 shows the spatial and temporal distributions of surface water $pCO_2$.", the spatial distribution of in-situ measurements or data from other source?

[Response]: Figure 3 shows the spatial distribution of in-situ $pCO_2$. In the revision, we have added more details as follows: "Figure 3 shows the spatial and temporal distributions of in situ sea surface $pCO_2$." (Line 147) .

line 157:   how the remote sensing-derived $pCO_2$ data were derived? which methods, what is the quality? and output from unpublished study should not be used.

[Response]: This dataset is an updated version based on Bai et al. (2015). Please refer to our response to Major Comment # 9 above. In the revision, we have changed this citation to "Yu et al. (2022)", and also added more details of this dataset as follows "Yu et al. (2022) subsequently used a non-linear regression method to develop a retrieval algorithm for seawater $pCO_2$ in the China Seas, and the RS-derived $pCO_2$ data from 2003-2018 were provided by the SatCO$_2$ platform (www.SatCO2.com). In the retrieval algorithm of Yu et al. (2022), the input parameters include sea surface temperature, chlorophyll-a concentration, remote sensing reflectance of three bands (Rrs412, 443, 488 nm), the temperature anomaly in the longitudinal direction, and the theoretical thermodynamic background $pCO_2$ under the corresponding SST. Although the RMSE associated with the RS-derived $pCO_2$ product was relatively large (21.1 μatm), it successfully showed major spatial patterns of the sea surface $pCO_2$ in the China Seas (Yu et al., 2022)." (Lines 81-86).

line 184-187: "Wang et al. (in preparation) found a relatively high differential between the....observed data", meaning of this super long sentence is not clear.

[Response]: Accepted. In the revision, we have modified this sentence as follows "For sea surface salinity (SSS) data, Wang et al. (2022) found relatively large differences between different open source SSS databases (i.e., multi-satellite fusion data from https://podaac.jpl.nasa.gov/; model data from https://climatedataguide.ucar.edu/; multidimensional covariance model data from https://resources.marine.copernicus.eu/) and the in situ SSS data." (Lines 186-189).

line 198 "$pCO_2$ filling method of", should explain the filling method here!

[Response]: Accepted. In the revision, we have modified the sentence in lines 208-209 as follows "Secondly, we filled missing $pCO_2$ measurements with the RS-derived $pCO_2$ data according to Fay et al. (2021) (see more details in Section 3.1)." , because the $pCO_2$ filling method is explained in section 3.1.

line 201: "$pCO_2$ reconstruction model"   $pCO_2$ reconstruction was used many times in the manuscript, but sea surface $pCO_2$ is not something one can reconstruct, it is a properties or variable of of the sea water, one can measure it ,describe it, retrieve its distribution, but not reconstruct $pCO_2$ itself.   So, please pay attention to the verb usage.

[Response]: Accepted. In the revision, we have changed "$pCO_2$ reconstruction model" to "$pCO_2$ retrieval algorithm".

References

Dye, A. W., Rastogi, B., Clemesha, R. E. S., Kim, J. B., Samelson, R. M., Still, C. J., & Williams, A. P.: Spatial patterns and trends of summertime low cloudiness for the Pacific Northwest, 1996–2017. Geophysical Research Letters, 47, e2020GL088121, 2020. https://doi.org/10.1029/2020GL088121

Levitus, S., Antonov, J. I., Boyer, T. P., Garcia, H. E., and Locarnini, R. A.: EOF analysis of upper ocean heat content, 1956–2003, Geophys. Res. Lett., 32, L18607, 2005. doi:10.1029/2005GL023606.

McMonigal, K., & Larson, S. M.: ENSO explains the link between Indian Ocean dipole and Meridional Ocean heat transport. Geophysical Research Letters, 49, e2021GL095796, 2022. https://doi.org/10.1029/2021GL095796

Wang, Z., Wang, G., Guo, X., Hu, J., and Dai, M. Reconstruction of High-Resolution Sea Surface Salinity over 2003–2020 in the South China Sea Using the Machine Learning Algorithm LightGBM Model. Remote. Sens., 14, 6147, 2022. https://doi.org/10.3390/rs14236147.

Yu, S., Song, Z., Bai, Y., and He, X.: Remote Sensing based Sea Surface partial pressure of $CO_2$ ($pCO_2$) in China Seas (2003-2019) (2.0). Zenodo, 2022. https://doi.org/10.5281/zenodo.7372479.